# Feasible Fusion: Constrained Joint Estimation under Structural Non-Overlap

Yuxi Du [1]   Zhiheng Zhang[✉][1][2]   Haoxuan Li [3]   Cong Fang [3]   Jixing Xu [4]   Zhen Peng [4]   Jiecheng Guo [4]

## Abstract

Causal inference in modern large-scale systems faces growing challenges, including high-dimensional covariates, multi-valued treatments, massive observational (OBS) data, and limited randomized controlled trial (RCT) samples due to cost constraints. We formalize treatment-induced structural non-overlap and show that, under this regime, commonly used weighted fusion methods provably fail to satisfy randomized identifying restrictions. To address this issue, we propose a *constrained joint estimation framework* that minimizes observational risk while enforcing causal validity through orthogonal experimental moment conditions. We further show that structural non-overlap creates a feasibility obstruction for moment enforcement in the original covariate space. We also derive a penalized primal–dual algorithm that jointly learns representations and predictors, and establish oracle inequalities decomposing error into overlap recovery, moment violation, and statistical terms. Extensive synthetic experiments demonstrate robust performance under varying degrees of non-overlap. A large-scale ride-hailing application shows that our method achieves substantial gains over existing baselines, matching the performance of models trained with significantly more RCT data.

## 1. Introduction

Modern empirical decision-making increasingly relies on two complementary sources of evidence: randomized controlled trials (RCTs), which provide high internal validity through randomized treatment assignment, and large-scale observational (OBS) logs, which capture rich heterogeneity at scale but may suffer from confounding and policy-induced selection effects. A central goal in contemporary causal inference is to combine these data sources so as to obtain estimators that are simultaneously *causally valid* and *statistically efficient*; see, e.g., Imbens & Rubin (2015), Degtiar & Rose (2023), and the recent review by Colnet et al. (2024). This goal is especially consequential in domains where small changes in decision rules can translate into large welfare or revenue impacts, such as online platforms, transportation marketplaces, and other large-scale allocation systems.

Within this broad agenda, we focus on *heterogeneous response estimation* under *structured treatments*. Rather than a binary intervention, many real systems deploy multi-valued, high-cardinality, or combinatorial treatment assignments (e.g., discretized policy parameters, bundles of operational actions, or constrained action sets), for which classical propensity-score ideas must be interpreted in a generalized form (Imai & Van Dyk, 2004). In such settings, the target object is naturally expressed as a family of conditional response functions $\mu_t(x) = \mathbb{E}[Y(t) \mid X = x]$, $t \in \mathcal{T}$, where $X$ denotes covariates, $t$ denotes a structured treatment, and $Y(t)$ is the potential outcome under $t$. A growing body of work studies how to leverage RCT information to "ground" or regularize estimates learned from observational logs (and conversely, how to improve the external validity of RCT findings using observational samples), under assumptions that make the two sources compatible (Degtiar & Rose, 2023; Colnet et al., 2024; Bareinboim & Pearl, 2016). At a methodological level, this literature connects to (i) generalizability/transportability analyses, (ii) debiasing or robustness-based fusion, and (iii) representation-learning approaches that attempt to mitigate distribution shift by learning balanced covariate embeddings (Johansson et al., 2016; Shalit et al., 2017; Shi et al., 2019).

Despite this progress, much of the existing theory and practice of data fusion relies—often implicitly—on some form of overlap or positivity: treatments of interest are assumed to occur with non-negligible probability across relevant covariate strata (Rosenbaum & Rubin, 1983). When overlap is weak but nonzero, standard remedies include trimming regions with extreme propensities, reweighting samples, or

---

[1]School of Statistics and Data Science, Shanghai University of Finance and Economics, Shanghai, China [2]Institute of Big Data Research, Shanghai University of Finance and Economics, Shanghai, China [3]State Key Lab of General AI, School of Intelligence Science and Technology, Peking University, Beijing, China [4]Didi Chuxing, Beijing, China. Correspondence to: Zhiheng Zhang <zhangzhiheng@mail.shufe.edu.cn>.

*Proceedings of the 43rd International Conference on Machine Learning*, Seoul, South Korea. PMLR 306, 2026. Copyright 2026 by the author(s).

tolerating extrapolation, each of which trades bias against variance and weakens finite-sample guarantees (CRUMP et al., 2009). However, in many modern platform environments with structured and constrained decision rules, the dominant difficulty is of a qualitatively different nature. Here, certain covariate–treatment combinations are not merely rare, but *structurally impossible* under the production policy and therefore never appear in observational logs. This absence is induced by the decision mechanism itself—such as hard constraints, safety rules, or operational policies—rather than by sampling variability or limited data. As a consequence, increasing the size of observational data or adjusting weights cannot recover these missing regions of support, rendering classical overlap-based remedies ineffective.

Consider a ride-hailing platform (e.g., Didi) that operates under supply constraints, operational rules, and safety/business restrictions. Treatments may encode combinations of dispatch actions, pricing multipliers, or policy parameters, and the deployed policy may deterministically exclude large subsets of the action set in particular contexts. Consequently, there can exist measurable regions of $(x, t)$ for which an experimental design assigns positive probability (because exploration is explicitly introduced), while the observational logging policy assigns probability zero. In such a regime, naive pooling—or any procedure that implicitly treats mixed RCT–OBS data as if generated from a single compatible design—faces a fundamental dilemma: either it *drifts* toward observationally optimal predictors that violate experimental identifying restrictions, or it *extrapolates* individual-level effects into regions unsupported by the observational policy. Crucially, this failure mode is shared across a broad family of approaches, including weighted-loss fusion and representation-learning pipelines that are trained without explicitly encoding experimental identification, because the missing support cannot be created by tuning weights or by balancing objectives alone (Johansson et al., 2016; Shalit et al., 2017).

Our contributions could be summarized as follows.

- We formalize *treatment-induced structural non-overlap* for structured treatments and show that it creates a fundamental support and feasibility obstruction for standard overlap-based and weighted fusion of randomized and observational data.

- We propose a constrained joint estimation framework that enforces randomization-induced experimental moment conditions while minimizing observational risk, together with a representation-learning scheme that restores approximate feasibility via distribution alignment.

- We establish theoretical separation results and oracle-

type risk bounds under structural non-overlap, and demonstrate substantial empirical gains in synthetic studies and a large-scale ride-hailing application where observational data are abundant but experiments are costly.

**Organization** Section 2 introduces the setup and the notion of treatment-induced structural non-overlap. Section 3 develops the constrained joint-estimation framework and presents the main theoretical results on feasibility recovery and risk bounds. Section 4 and 5 reports synthetic and real-data experiments.

In contrast to prior work, we adopt a constrained joint estimation paradigm that explicitly separates causal validity from observational efficiency. Rather than soft fusion via weighted empirical risks, we treat experimental randomization as a source of moment constraints that must be satisfied exactly in the population. Moreover, unlike two-stage pipelines that first learn representations and subsequently enforce causal restrictions, our framework learns representations jointly with the experimental constraints, recognizing feasibility restoration as an inherently causal—rather than purely predictive—objective. This joint, constraint-driven view enables identification and estimation in regimes where existing integrative methods provably break down.

## 2. Problem Setup

We consider units indexed by $i = 1, \ldots, n$ with covariates $X \in \mathcal{X} \subset \mathbb{R}^d$, structured treatments $T \in \mathcal{T}$, and outcomes $Y \in \mathbb{R}$. The treatment space $\mathcal{T}$ may be multi-valued or combinatorial, representing policy rules or action bundles rather than a binary intervention. Potential outcomes $\{Y(t) : t \in \mathcal{T}\}$ are defined in the Rubin causal model.

We observe two data sources generated under distinct assignment mechanisms: **(i) RCT data** from $P_r(X, T, Y)$, where treatment is randomized conditional on $X$, and **(ii) observational (OBS) data** from $P_o(X, T, Y)$, where treatment follows a production policy that may depend on $X$ and unobserved factors. The distributions $P_r$ and $P_o$ may differ in both covariate marginals and treatment assignment.

Our target estimand is the conditional response function

$$\mu_t(x) = \mathbb{E}[Y(t) \mid X = x], \qquad t \in \mathcal{T}, \qquad (1)$$

which captures heterogeneous, individual-level causal effects under structured treatments.

Classical identification assumes overlap, $P_o(T = t \mid X = x) > 0$ for all $(x, t)$. In contrast, under constrained or rule-based assignment, overlap violations are *structural*: certain covariate–treatment pairs are never generated by the observational policy, irrespective of sample size.

Next, we will provide two specific definitions of structural non overlap and explain why Definition 2.1 can only be solved by extrapolation, inevitably introducing model spcification bias, while Definition 2.2 can recover representations through joint training, thus becoming the research topic of this article.

**Definition 2.1** (Marginal (irreducible) structural non-overlap). We say that the observational and randomized distributions exhibit *marginal structural non-overlap* if there exists a measurable set $A \subseteq \mathcal{T}$ such that

$$P_r(T \in A) > 0 \qquad \text{and} \qquad P_o(T \in A) = 0.$$

In this case, the support mismatch already occurs at the marginal level of the treatment, independently of the covariates $X$.

**Definition 2.2** (Conditional (recoverable) structural non-overlap). We say that the observational and randomized distributions exhibit *conditional (recoverable) structural non-overlap* if the treatment has overlapping marginal support under $P_r$ and $P_o$, i.e.,

$$\forall A \subseteq \mathcal{T}, \quad P_o(T \in A) = 0 \ \Rightarrow \ P_r(T \in A) = 0,$$

but there exists a measurable set $S \subseteq \mathcal{X} \times \mathcal{T}$ such that

$$P_r(S) > 0 \quad \text{and} \quad P_o(S) = 0.$$

That is, the marginal support of the treatment is aligned under $P_r$ and $P_o$, while the joint support mismatch arises conditionally on $X$.

*Remark* 2.3 (Recoverable versus irreducible non-overlap). Definitions 2.1 and 2.2 distinguish recoverable and irreducible sources of structural non-overlap. Under marginal structural non-overlap, the mismatch occurs in the marginal treatment support and is irreducible: since representations only transform covariates, no $\phi$ can eliminate this mismatch (Theorem B.10). Extrapolation-based approaches implicitly operate in this regime by imposing strong outcome-model assumptions to predict unsupported treatment effects; however, such methods do not restore identification via randomized moment validity and may incur persistent misspecification bias.

In contrast to irreducible marginal non-overlap, conditional structural non-overlap admits that a representation $\phi(X)$ can aggregate regions with zero observational support into equivalence classes with positive support under the observational policy, thereby restoring overlap in the induced space. This enables approximate randomized moment validity without extrapolation and clarifies the distinction between recoverable and irreducible non-overlap underlying the feasibility guarantee in Corollary 3.7 and the impossibility result in Theorem B.10.

**Representation learning for feasibility recovery.** Motivated by this insight, we introduce a representation map

$$\phi : \mathcal{X} \to \mathcal{Z}, \tag{2}$$

whose role is to *endogenously restore feasibility* of causal identification by balancing overlap recovery against outcome-relevant information preservation, rather than performing generic dimensionality reduction, invariance, or distributional balancing.

**Definition 2.4** (Representation Property for Joint Estimation). A representation map $\phi$ is said to satisfy the *representation property for joint estimation* if it fulfills both the overlap recovery and information preservation conditions stated below.

**Proposition 2.5** (Overlap Recovery). *Let $\mathcal{D}$ be a class of measurable test functions on $\mathcal{Z} \times \mathcal{T}$ and let $\mathrm{IPM}_{\mathcal{D}}(\cdot, \cdot)$ denote the associated integral probability metric. The representation $\phi$ is said to recover overlap if there exists $\varepsilon_{\mathrm{ov}} > 0$ such that*

$$\mathrm{IPM}_{\mathcal{D}}(P_r(\phi(X), T), \ P_o(\phi(X), T)) \ \leq \ \varepsilon_{\mathrm{ov}}, \tag{3}$$

*even though the original covariate space may exhibit violated overlap in the sense that* $\mathrm{supp}(X \mid T = t_1) \cap \mathrm{supp}(X \mid T = t_0) = \varnothing$ *for some $t_0, t_1 \in \mathcal{T}$.*

**Proposition 2.6** (Outcome-Relevant Information Preservation). *Let $\mathcal{F}$ be a hypothesis class of measurable functions on $\mathcal{X}$. The representation $\phi$ is said to preserve outcome-relevant information if there exists $\varepsilon_{\mathrm{info}} > 0$ such that*

$$\inf_{f \in \mathcal{F}} \mathbb{E}\big[(Y - f(\phi(X)))^2\big] \ \leq \ \inf_{g \in \mathcal{F}} \mathbb{E}\big[(Y - g(X))^2\big] + \varepsilon_{\mathrm{info}}, \tag{4}$$

*that is, conditioning on $\phi(X)$ incurs at most an $\varepsilon_{\mathrm{info}}$ excess prediction risk relative to conditioning on the full covariate vector $X$.*

**Technical challenge.** Propositions 2.5 and 2.6 impose competing requirements on the representation $\phi$. Overlap recovery Eq (3) favors compressive mappings that reduce discrepancy between $P_r$ and $P_o$, whereas information preservation Eq (4) requires retaining outcome-relevant structure. These objectives are inherently in tension (see Theorem B.15), especially in high-dimensional settings with structured treatments, and cannot be simultaneously ensured by representations learned independently of causal constraints.

*Remark* 2.7. Most existing representation-learning approaches for causal inference implicitly assume that a representation satisfying both Eq (3) and Eq (4) exists and can be obtained through balancing or invariance-based objectives. Under treatment-induced structural non-overlap, this assumption is neither verifiable from data nor robust to model misspecification. In contrast, our framework treats

representation learning as an endogenous component of causal estimation, explicitly quantifying feasibility through $\varepsilon_{\text{ov}}$ and $\varepsilon_{\text{info}}$ and analyzing how these quantities propagate into estimation error in subsequent sections.

# 3. Method: From Oracle Moment-based Joint Estimation to the Representation Space

This section develops a constrained joint-estimation framework for fusing observational logs with randomized data in the presence of *Conditional (recoverable) structural non-overlap* (Definition 2.2). We proceed in four steps. Section 3.1 derives experimental moment conditions and explains why they constrain *causal components* while leaving nuisance structure unconstrained. Section 3.2 formulates data fusion as a constrained program and establishes a separation from weighted-loss fusion. Section 3.3 shows how structural non-overlap manifests as a *feasibility obstruction* in the original covariate space and motivates representation learning as feasibility recovery. Section 3.4 presents a primal–dual algorithm and a penalized objective suitable for large-scale training, and Section 3.5 states an oracle inequality that decomposes error into overlap, moment, and statistical terms.

## 3.1. Experimental moments as identifying restrictions

For clarity of exposition, assume the treatment space is finite, $\mathcal{T} = \{0, 1, \ldots, K\}$, where $0$ denotes a reference treatment (e.g., control) and $k \in \{1, \ldots, K\}$ denote active treatments. Write the one-hot encoding $D = (D_0, \ldots, D_K)$ with $D_k = \mathbf{1}\{T = k\}$ and $\sum_{k=0}^{K} D_k = 1$. In an RCT, the assignment probabilities $p_k := \mathbb{P}_r(T = k)$ $(k = 0, \ldots, K)$ are known by design (more generally, under stratified randomization one may take $p_k(x) = \mathbb{P}_r(T = k \mid X = x)$; all statements below extend by replacing $p_k$ with $p_k(X)$).

Two simple identities motivate our construction. First, the observed outcome admits the potential-outcome decomposition $Y = \sum_{k=0}^{K} D_k Y(k) = Y(0) + \sum_{k=1}^{K} D_k (Y(k) - Y(0))$, which is not an assumption, but a re-expression of $Y = \sum_k D_k Y(k)$. Second, any measurable regression function $m(x, t)$ can be written as $m(x, t) = u(x) + \sum_{k=1}^{K} \mathbf{1}\{t = k\} h_k(x), u(x) := m(x, 0), h_k(x) := m(x, k) - m(x, 0)$. Thus, the pair $(u, \{h_k\}_{k=1}^{K})$ separates a *baseline* term $u$ (common across treatments) from *treatment-specific* increments $h_k$.

Define, for each $k \in \{1, \ldots, K\}$, the moment map

$$\psi_k(Y, T, X; m) := (D_k - p_k)(Y - m(X, T)), \quad (5)$$

where $\psi := (\psi_1, \ldots, \psi_K)^\top$. Under randomization, the conditional mean function $m_r(x, t) := \mathbb{E}_{P_r}[Y \mid X = x, T =$

$t]$ satisfies

$$\|\mathbb{E}_{P_r}[\psi(Y, T, X; m_r)]\| = 0. \quad (6)$$

We treat Eq (6) as an identifying restriction: any causally admissible predictor should satisfy the moment conditions (exactly in the population, approximately in finite samples).

**Lemma 3.1** (Baseline invariance of experimental moments). *Let* $u : \mathcal{X} \to \mathbb{R}$ *be measurable. Under RCT randomization with known assignment probabilities* $p_k$, $\mathbb{E}_{P_r}[(D_k - p_k) u(X)] = 0$, $\forall k \in \{1, \ldots, K\}$. *Consequently, for any* $m$ *parameterized as above, the moment* $\mathbb{E}_{P_r}[\psi_k(Y, T, X; m)]$ *depends on* $m$ *only through the treatment-specific components* $\{h_k\}$.

Expanding the moment using above equations yields

$$\mathbb{E}_{P_r}[(D_k - p_k)(Y - m(X, T))] \quad (7)$$

$$= \underbrace{\mathbb{E}_{P_r}[(D_k - p_k)(Y(0) - u(X))]}_{\text{baseline / nuisance component}}$$

$$+ \underbrace{\mathbb{E}_{P_r}\left[(D_k - p_k) \sum_{i=1}^{K} D_i (Y(i) - Y(0) - h_i(X))\right]}_{\text{causal component}}.$$

$$(8)$$

By Lemma 3.1, the baseline term vanishes under randomization, leaving only the causal component. This is the key reason to use moments rather than an RCT squared loss: the moment conditions constrain the *causal increments* $\{h_k\}$ while remaining agnostic about baseline outcome structure. Importantly, Eq (6) is *not* equivalent to fitting $m$ by minimizing an RCT squared loss, which couples baseline and causal components and may enforce spurious agreement between RCT and OBS outcome models. A common misconception is that "giving the RCT a large weight" in a weighted regression objective should recover causal validity. Theorem 3.4 below formalizes why this intuition can fail under structural non-overlap: weighted-loss fusion may remain bounded away from satisfying Eq (6), regardless of tuning.(In practice, RCT samples are often scarce, so pushing $\alpha$ into the RCT-dominated regime can yield high-variance and unstable estimates.)

## 3.2. Fusion as a constrained program: optimality and separation

Let $\mathcal{M}$ be a model class of measurable predictors $m : \mathcal{X} \times \mathcal{T} \to \mathbb{R}$. Define the observational risk $\mathcal{R}_o(m) := \mathbb{E}_{P_o}[(Y - m(X, T))^2]$. Guided by the complementary roles of the two data sources, we propose the constrained program

$$\min_{m \in \mathcal{M}} \mathcal{R}_o(m) \quad \text{s.t.} \quad \mathbb{E}_{P_r}[\psi(Y, T, X; m)] = 0, \quad (9)$$

where $\psi$ is defined in Eq (5). The objective exploits the abundance of observational data to learn high-dimensional

nuisance structure, while the constraint enforces causal validity through experimental randomization.This distinction is particularly important in modern applications, where randomized traffic is scarce and the number of treatment arms is large. In such regimes, RCT-only estimation suffers from severe small-sample effects, leading to high-variance and unstable estimates of treatment effects, especially at the individual level. The constrained program mitigates this issue by decoupling nuisance learning from causal calibration: observational data are used to estimate complex baseline and representation components, while randomized data are used only to enforce causal validity.

**Assumption 3.2** (Constraint feasibility and regularity). The feasible set $\mathcal{F} := \{m \in \mathcal{M} : \mathbb{E}_{P_r}[\psi(Y, T, X; m)] = 0\} \neq \emptyset$. Moreover, $\mathcal{M}$ is convex and closed, and the mapping $m \mapsto \mathbb{E}_{P_r}[\psi(Y, T, X; m)]$ is continuous.

Under Assumption 3.2, any solution $m^\star$ to Eq (9) satisfies $\mathcal{R}_o(m^\star) = \inf_{m \in \mathcal{F}} \mathcal{R}_o(m)$. Equivalently, Eq (9) returns the best observational predictor among all models that satisfy the randomization-induced moment restrictions.

A widely used alternative is to combine RCT and OBS objectives via a weighted squared loss:

$$m_\alpha \in \arg\min_{m \in \mathcal{M}} \left\{ \mathcal{R}_o(m) + \alpha \, \mathcal{R}_r(m) \right\}, \qquad (10)$$

where $\mathcal{R}_r(m) := \mathbb{E}_{P_r}\left[(Y - m(X, T))^2\right]$ with tuning parameter $\alpha \geq 0$. Unlike Eq (9), Eq (10) does *not* encode experimental identification, and may therefore trade off causal validity against predictive fit. The remainder of Section 3.2 shows that, in typical regimes, weighted-loss fusion cannot enforce the experimental moment conditions; moreover, aggressive tuning effectively reduces to an RCT-only estimator, yielding high-variance and unstable treatment-effect estimates.

**Assumption 3.3** (When weighted-loss fusion cannot satisfy experimental moments). **(i) Observational fit gap induced by causal validity.** There exists $\delta_o > 0$ such that

$$\inf_{m \in \mathcal{F}} \mathcal{R}_o(m) \geq \inf_{m \in \mathcal{M}} \mathcal{R}_o(m) + \delta_o, \qquad (11)$$

i.e., enforcing the randomized moment restrictions incurs a strictly positive increase in observational prediction risk.

**(ii) Bounded range of the RCT risk.** Define

$$B_r := \sup_{m \in \mathcal{M}} \mathcal{R}_r(m) - \inf_{m \in \mathcal{M}} \mathcal{R}_r(m), \qquad 0 < B_r < \infty. \qquad (12)$$

**(iii) Non-degenerate fusion regime and regularity.** Consider weighted-loss fusion with $\alpha \in [0, \bar{\alpha}]$, where

$$\bar{\alpha} := \frac{\delta_o}{2B_r}. \qquad (13)$$

For each $\alpha \in [0, \bar{\alpha}]$, the weighted objective Eq (10) admits at least one minimizer $m_\alpha \in \mathcal{M}$, and the set $\{m_\alpha : \alpha \in [0, \bar{\alpha}]\}$ is compact in $L_2(P_r)$. Moreover, the map $m \mapsto \mathbb{E}_{P_r}[\psi(Y, T, X; m)]$ is continuous on $\mathcal{M}$.

**Theorem 3.4** (Separation from weighted-loss fusion in the fusion regime). *Under Assumptions 3.2 and 3.3, the constrained solution $m^\star$ to Eq (9) satisfies the experimental moments exactly:*

$$\|\mathbb{E}_{P_r}[\psi(Y, T, X; m^\star)]\| = 0.$$

*Moreover, there exists a constant $c_0 > 0$ such that every weighted-loss minimizer $m_\alpha$ of Eq (10) with $\alpha \in [0, \bar{\alpha}]$ violates the moments by at least $c_0$:*

$$\inf_{\alpha \in [0, \bar{\alpha}]} \|\mathbb{E}_{P_r}[\psi(Y, T, X; m_\alpha)]\| \geq c_0.$$

*Consequently, weighted-loss fusion with non-negligible observational weight ($\alpha \leq \bar{\alpha}$) is fundamentally incompatible with exact satisfaction of the randomized moments, which can only be achieved by taking $\alpha > \bar{\alpha}$.*

Furthermore, Corollary B.11 controls the weighted-solution path by showing that $\|g(m_\alpha)\|$ can decrease at most linearly in $\alpha$. This rules out an isolated "sweet spot" at small $\alpha$ where feasibility would suddenly hold (see Appendix C).

We also prove that weighted-loss fusion coincides with the constrained estimator only in *degenerate* settings(Assumption 3.3 fails), requiring strong alignment between observational risk, RCT risk, and the identifying moments (e.g., shared minimizers, exact KKT alignment). (see Appendix D). Such conditions are highly idealized and are generically violated under structural non-overlap, where exact moment satisfaction incurs a positive observational risk penalty. Consequently, weighted-loss fusion exhibits an irreducible moment violation in non-degenerate regimes.

This analysis above motivates the constrained formulation in Section 3.3, which enforces causal validity explicitly and uses observational data only within the causally admissible set, rather than relying on scalar reweighting to implicitly satisfy identifying restrictions.

### 3.3. Structural non-overlap as a feasibility obstruction and the role of representation

The constrained formulation in Section 3.2 assumes feasibility of the identifying moment constraints within the function class $\mathcal{M}$ (Assumption 3.2). Under conditional non-overlap (Definition 2.2), however, this assumption may fail: no predictor in $\mathcal{M}$ may simultaneously satisfy the experimental moments and achieve adequate observational fit in the original covariate space.

Rather than imposing feasibility *a priori*, we view it as a property that may be *recovered or approximated* through

representation learning. Accordingly, this section characterizes when representation learning can restore feasibility and when causal validity remains provably unattainable.

**Definition 3.5** (Feasibility gap). Let $\psi$ be the moment map in Eq ($5$). The *feasibility gap* of $\mathcal{M}$ in the original covariate space is $\mathcal{F}_X(\mathcal{M}) := \inf_{m \in \mathcal{M}} \|\mathbb{E}_{P_r}[\psi(Y, T, X; m)]\|$. For a representation $\phi : \mathcal{X} \rightarrow \mathcal{Z}$ and induced class $\mathcal{M}_\phi := \{(x, t) \mapsto m(\phi(x), t) : m \in \mathcal{M}\}$, define the latent-space feasibility gap $\mathcal{F}_\phi(\mathcal{M}) := \inf_{m \in \mathcal{M}_\phi} \|\mathbb{E}_{P_r}[\psi(Y, T, \phi(X); m)]\|$.

A strictly positive $\mathcal{F}_X(\mathcal{M})$ means that exact moment satisfaction is unattainable in the original space; feasibility must be recovered by altering the hypothesis class, which we do via representation learning.

**Theorem 3.6** (Explicit feasibility gap in the original space). *Let $\mathcal{M}$ be a model class of measurable predictors $m :$ $\mathcal{X} \times \mathcal{T} \rightarrow \mathbb{R}$ Under Assumption 2.2 (so that there exists a measurable $S \subseteq \mathcal{X} \times \mathcal{T}$ with $P_r(S) > 0$ and $P_o(S) = 0$), Assumptions B.4–B.6, and Lemma C.1, the feasibility gap in the original covariate space is strictly positive:*

$$\mathcal{F}_X(\mathcal{M}) = \inf_{m \in \mathcal{M}} \left\| \mathbb{E}_{P_r}[\psi(Y, T, X; m)] \right\| \geq c_0,$$

*where*

$$c_0 := \underline{p}(1 - \bar{p}) P_r(S) \delta > 0,$$

*and $(\underline{p}, \bar{p})$ and $\delta$ are the constants in Assumptions B.6 and B.5, respectively.*

Theorem 3.6 formalizes a feasibility *obstruction* in the original covariate space: under conditional structural non-overlap and mild misspecification, the identifying moments incur a strictly positive residual $c_0$, i.e., exact feasibility is unattainable within $\mathcal{M}$ on $X$. This suggests that enforcing the randomized moments necessarily requires altering the effective hypothesis class.

Motivated by Theorem 3.6, we pursue feasibility recovery by *changing the space* in which the identifying moments are enforced. Rather than fixing a representation *a priori*, we consider representations $\phi$ as part of the optimization, each inducing a hypothesis class $\mathcal{M}_\phi$ with a potentially different feasibility profile. Our goal is therefore to identify representations under which the randomized moments become (approximately) satisfiable, while simultaneously minimizing observational risk.

This perspective leads to the following latent constrained program:

$$\min_{\phi \in \Phi} \min_{m \in \mathcal{M}_\phi} \mathcal{R}_o(m) \quad \text{s.t.} \quad \mathbb{E}_{P_r}[\psi(Y, T, \phi(X); m)] = 0,$$

$$(14)$$

which should be interpreted as a *joint* optimization over representations and predictors, rather than a sequential two-

stage procedure. This formulation directly bridges the raw-space infeasibility established in Theorem 3.6 with the strict feasibility improvement guaranteed next in Corollary 3.7.

**Corollary 3.7** (Strict improvement via representation learning). *Let $\mathcal{M}$ be a class of measurable predictors $m :$ $\mathcal{X} \times \mathcal{T} \rightarrow \mathbb{R}$. For any representation $\phi : \mathcal{X} \rightarrow \mathcal{Z}$, define the induced class*

$$\mathcal{M}_\phi := \{m_\theta(\phi(\cdot), \cdot) : \theta \in \Theta\},$$

*and the randomized moment map*

$$g_\phi(m) := \mathbb{E}_{P_r}[\psi(Y, T, \phi(X); m)] \in \mathbb{R}^K.$$

*Assume the raw-space obstruction $\mathcal{F}_X(\mathcal{M}) \geq c_0 > 0$. Further assume that feasibility is controlled by overlap in the induced space: there exists $c_1 > 0$ such that, for all $\phi \in \Phi$,*

$$\mathcal{F}_\phi(\mathcal{M}) \leq c_1 \varepsilon_{ov}(\phi).$$

*Under Assumptions B.3 and B.2, whenever $\mathcal{F}_\phi \neq \emptyset$, there exist constants $\mu_o > 0$ and $L_g > 0$ such that, for all $m \in \mathcal{M}_\phi$,*

$$\mathcal{R}_o(m) - \mathcal{R}_{o,\phi}^{\text{feas}} \geq \mu_o \operatorname{dist}(m, \mathcal{F}_\phi)^2,$$
$$\|g_\phi(m)\| \leq L_g \operatorname{dist}(m, \mathcal{F}_\phi),$$
$$(15)$$

*where*

$$\mathcal{R}_{o,\phi}^{\text{feas}} := \inf_{m \in \mathcal{F}_\phi} \mathcal{R}_o(m).$$

*Let $\phi^\star$ be returned by the latent constrained program in Eq. ($14$), and assume $\mathcal{F}_{\phi^\star} \neq \emptyset$. If $\varepsilon_{ov}(\phi^\star) < c_0/c_1$, then*

$$\mathcal{F}_{\phi^\star}(\mathcal{M}) \leq c_1 \varepsilon_{ov}(\phi^\star)$$
$$< c_0 \leq \mathcal{F}_X(\mathcal{M}),$$
$$(16)$$

*so feasibility is strictly improved. In addition,*

$$\inf_{m \in \mathcal{M}_{\phi^\star}} \left\{ \mathcal{R}_o(m) - \mathcal{R}_{o,\phi^\star}^{\text{feas}} \right\}$$
$$\geq \frac{\mu_o}{L_g^2} \left( \mathcal{F}_{\phi^\star}(\mathcal{M}) \right)^2.$$
$$(17)$$

We characterize the dependence of $c_1$ on the discriminator class $\mathcal{D}$ (via an embedding/dominance constant) in Appendix F.

Corollary 3.7 formalizes the central intuition behind representation learning under *conditional (recoverable) structural non-overlap* (Definition 2.2). In this regime, although certain actions are never taken in specific contexts under the observational policy (i.e., $P_o(x, t) = 0$ while $P_r(x, t) > 0$), the marginal support of treatments remains aligned between observational and randomized data. Consequently, there exists an appropriate representation $\phi^\star$ that coarsens or reparameterizes the covariate space to recover joint support, rendering the identifying moments approximately feasible

and leading to strict improvements in both feasibility and observational risk.

However,even within the recoverable regime, feasibility recovery is not without cost. Although representation learning avoids genuine extrapolation, the attainable improvement is jointly constrained by residual distribution mismatch between observational and randomized data and by the amount of outcome-relevant information retained. Theorem B.15 formalizes this limitation through an explicit feasibility–information trade-off, showing that improving one without controlling the other is insufficient.

This perspective also highlights a common pitfall: learning representations independently (e.g., for prediction or balance) and enforcing causal restrictions afterward. As feasibility restoration is inherently a causal objective, a representation may preserve predictive information yet fail to reduce $\varepsilon_{\mathrm{ov}}$, leaving the feasible set $\mathcal{F}_\phi(\mathcal{M})$ large. Our approach therefore learns $\phi$ jointly with the identifying moment constraints, ensuring that overlap recovery and causal feasibility are optimized *together*, as illustrated in the next section.

Finally, feasibility recovery may be fundamentally impossible in certain regimes. As shown by the minimax lower bound in Appendix D, under *irreducible* structural non-overlap (Definition 2.1), no representation can reduce the feasibility gap below a constant multiple of the residual overlap discrepancy. Such regimes require genuine extrapolation beyond observational support and therefore fall outside the scope of this work, which focuses on feasibility recovery via joint representation learning rather than extrapolation.

### 3.4. Augmented Lagrangian and a primal–dual algorithm

In spired by Corollary 3.7 above, we introduce augmented-Lagrangian primal–dual scheme, which enforces the identifying moments while promoting overlap in the induced space in solving Eq (14).

Define the (population) randomized moment residual

$$g(\theta, \phi) := \mathbb{E}_{P_r}\big[\psi(Y, T, \phi(X); m_\theta)\big] \in \mathbb{R}^K.$$

Let $R_o(\theta, \phi) := \mathbb{E}_{P_o}\big[(Y - m_\theta(\phi(X), T))^2\big]$ be the population observational risk. We minimize the augmented Lagrangian

$$\mathcal{L}_\rho(\theta, \phi, \nu) := \mathcal{R}_o(\theta, \phi) + \lambda\, \varepsilon_{\mathrm{ov}}(\phi)$$
$$+ \langle \nu,\, g(\theta, \phi) \rangle + \frac{\rho}{2}\|g(\theta, \phi)\|_2^2, \quad (18)$$

where $\nu \in \mathbb{R}^K$ is the dual variable, $\rho > 0$ is the augmentation parameter, and $\lambda \geq 0$ controls overlap regularization.

We employ a stochastic primal–dual (PD) optimization scheme, alternating gradient descent on the primal vari-

ables $(\theta, \phi)$ and gradient ascent on the dual variable $\nu$ using minibatch estimates. Algorithm 1 provides the complete update procedure. Appendix E proves PD is more stable than pure quadratic-penalty methods.

### 3.5. Oracle inequality and what it implies

We now connect algorithmic choice (provided in Section 3.4) to statistical performance: the following oracle inequality quantifies how moment violation, residual non-overlap, finite-sample fluctuations, and optimization suboptimality translate into excess observational risk.

**Theorem 3.8** (Oracle bound for a feasibility-augmented empirical program)**.** *Let $\widehat{R}_o(\theta, \phi)$ and $\widehat{g}(\theta, \phi)$ be empirical estimates based on $n_o$ observational and $n_r$ randomized samples. Assume Assumptions B.7 and B.8. Define the feasibility-augmented objectives*

$$Q(\theta, \phi) := R_o(\theta, \phi) + \frac{\mu_o}{L_g^2}\Big(\|g(\theta, \phi)\| + c_{\mathrm{ov}}\,\varepsilon_{\mathrm{ov}}(\phi)\Big)^2,$$

$$\widehat{Q}(\theta, \phi) := \widehat{R}_o(\theta, \phi) + \frac{\mu_o}{L_g^2}\Big(\|\widehat{g}(\theta, \phi)\| + c_{\mathrm{ov}}\,\varepsilon_{\mathrm{ov}}(\phi)\Big)^2.$$

*Let $(\hat{\theta}, \hat{\phi})$ be an $\eta$-approximate minimizer of $\widehat{Q}$, i.e., $\widehat{Q}(\hat{\theta}, \hat{\phi}) \leq \inf_{\theta, \phi} \widehat{Q}(\theta, \phi) + \eta$. Let $(\theta^\star, \phi^\star)$ be any oracle pair minimizing $Q$ (equivalently, any minimizer of $\inf_{\theta, \phi} Q(\theta, \phi)$). Then with probability at least $1 - \delta$,*

$$R_o(\hat{\theta}, \hat{\phi}) \leq R_o(\theta^\star, \phi^\star) + \frac{\mu_o}{L_g^2}\Big(\|g(\hat{\theta}, \hat{\phi})\| + c_{\mathrm{ov}}\,\varepsilon_{\mathrm{ov}}(\hat{\phi})\Big)^2$$
$$+ C\,\mathrm{Stat}(n_o, n_r) + \eta, \quad (19)$$

*for a universal constant $C > 0$.*

Theorem 3.8 establishes a near-oracle excess-risk bound for observational prediction. Specifically, Eq (19) decomposes the excess risk into three terms: a quadratic *feasibility term* $\frac{\mu_o}{L_g^2}\big(\|g(\hat{\theta}, \hat{\phi})\| + c_{\mathrm{ov}}\,\varepsilon_{\mathrm{ov}}(\hat{\phi})\big)^2$ capturing residual moment violation and overlap mismatch, a statistical error term $C \cdot \mathrm{Stat}(n_o, n_r)$,and the optimization suboptimality $\eta$.

Crucially, the theorem explains why joint primal–dual training is necessary for achieving near-oracle risk. Among these error sources, the feasibility residual $\|g(\hat{\theta}, \hat{\phi})\|$ cannot, in general, be driven to zero by methods that only reweight losses or fix the representation. In contrast, a primal–dual (augmented Lagrangian) procedure directly enforces $g(\theta, \phi) = 0$, and under controlled overlap and statistical error, approximate saddle-point optimality implies $\|g(\hat{\theta}, \hat{\phi})\| \to 0$ (cf. Proposition B.14).

## 4. Synthetic Experiment

This section evaluates the proposed method by first comparing predictive accuracy (MSE) and treatment-effect ranking

| Method | large_overlap | | moderate_overlap | | minimal_overlap | |
|---|---|---|---|---|---|---|
| | Qini (Mean±Std) | MSE (Mean±Std) | Qini (Mean±Std) | MSE (Mean±Std) | Qini (Mean±Std) | MSE (Mean±Std) |
| Own Method (Dual Loss + Wasserstein) | 0.5284±0.1336 | 0.9474±0.1551 | 0.5499±0.1202 | 1.0221±0.1116 | 0.5137±0.0744 | 0.7961±0.0945 |
| Own Method (Dual Loss Only) | 0.1670±0.2077 | 0.6906±0.2788 | 0.0942±0.2773 | 0.7517±0.3043 | 0.0254±0.1076 | 0.6295±0.2144 |
| Own Method (Wasserstein Only) | 0.1224±0.2217 | 0.7433±0.3030 | 0.1290±0.2336 | 0.7554±0.2517 | 0.0057±0.1450 | 0.5944±0.1601 |
| T-Learner | 0.4919±0.1815 | 0.8417±0.3851 | 0.5227±0.1810 | 0.7393±0.3602 | 0.0951±0.1818 | 0.9844±0.4109 |
| TARNet | 0.0370±0.4010 | 1.1621±0.5693 | 0.1620±0.3294 | 0.8011±0.2787 | 0.0196±0.1728 | 0.8409±0.3920 |
| CFRNet | 0.0688±0.3555 | 1.0916±0.6351 | 0.1783±0.3664 | 0.8858±0.4217 | -0.0229±0.1741 | 0.7580±0.2997 |
| DragonNet | 0.1194±0.4018 | 0.9785±0.5940 | 0.1701±0.3106 | 1.0110±0.6626 | 0.0379±0.1966 | 0.7777±0.2997 |
| Experimental Grounding | 0.2697±0.1900 | 6.1155±1.6609 | 0.2282±0.1596 | 7.9774±2.2844 | 0.1386±0.1009 | 18.5910±11.5983 |
| Integrative R-Learner | 0.3104±0.1916 | 2.0738±0.5419 | 0.3037±0.1703 | 2.7009±0.5453 | 0.2478±0.0910 | 4.1239±1.6499 |
| RCT-only R-Learner | 0.2516±0.2027 | 3.1449±0.7694 | 0.2719±0.1756 | 3.6738±0.7417 | 0.2094±0.0935 | 5.8326±2.8207 |
| Yang2024 Integrative HTE | 0.2762±0.2727 | 0.8671±0.3350 | 0.1821±0.3125 | 0.8716±0.3333 | 0.1554±0.3307 | 0.7317±0.2409 |
| Yang2022 Elastic Integrative | 0.4342±0.2596 | 0.6679±0.2892 | 0.1594±0.3327 | 0.9108±0.4571 | 0.3414±0.1716 | 0.4254±0.0875 |

*Table 1.* Performance comparison under different overlap conditions.

(Qini) with baseline approaches, followed by robustness and sensitivity analyses under severe Conditional non-overlap (see Defination 2.2).

### 4.1. Data Generation Process And Joint Mismatch

We compare the proposed approach with a set of representative baseline methods (see Appendix G.5) for integrating randomized controlled trial (RCT) and observational (OBS) data. These methods reflect the evolution of data-fusion strategies, ranging from early structurally grounded approaches to modern, largely non-structural neural estimators.

We construct a synthetic dataset that combines RCT and OBS samples to reflect realistic challenges arising from conditional structural non-overlap in high-dimensional covariate spaces. The data generation process is described in Appendix G, and additional visualizations illustrating joint distribution mismatch are provided in Appendix G.4.

### 4.2. Performance across datasets with varying overlap

We construct three types of datasets exhibiting different levels of joint distribution mismatch. Each dataset contains $n = 5,000$ samples. For each overlap setting, all methods are evaluated on 30 independently generated datasets, and we report the mean and standard deviation of the Qini coefficient and mean squared error (MSE) (see Table 1).

Across all overlap regimes, the proposed method consistently attains the highest Qini scores with the lowest variability, while also achieving near-minimal MSE in both mean and variance. Notably, its performance remains stable as overlap deteriorates, whereas competing methods exhibit substantial degradation when moving from large- to small-overlap settings.

To provide empirical evidence for the theoretical mechanisms underlying Corollary 3.7 and Assumption 3.3, we conduct additional simulations under varying degrees of conditional structural non-overlap. Specifically, these experiments are designed to examine two central theoretical

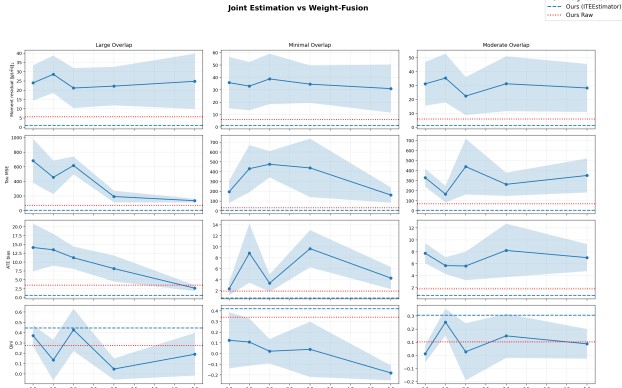

*Figure 1.* Feasibility recovery and downstream performance under varying degrees of conditional structural non-overlap.

claims: (i) whether representation learning can effectively recover feasibility under conditional non-overlap, and (ii) whether weighted-loss fusion remains unable to satisfy randomized moment conditions even when the weighting parameter $\alpha$ is extensively tuned.

As shown in Figure 1, the proposed joint framework with representation learning achieves the strongest feasibility recovery, followed by the constrained-only variant, while weighted-loss fusion exhibits the largest feasibility gap across a broad range of $\alpha$. The same ordering is observed for Qini and AUUC, supporting the theoretical link between feasibility recovery and treatment-effect estimation quality.

### 4.3. Robustness under Severe Conditional Non-overlap

To assess robustness in settings closer to real-world scenarios, we generate 50 datasets exhibiting varying degrees of severe conditional non-overlap. In spite of different levels, all datasets in this setting suffer from *Severe* conditional non-overlap. Figure 2 summarizes the performance of all methods in terms of mean squared error (MSE), a feasibility-related metric (a variation of $\|g(\theta, \phi)\|$), and the induced IPM distance.

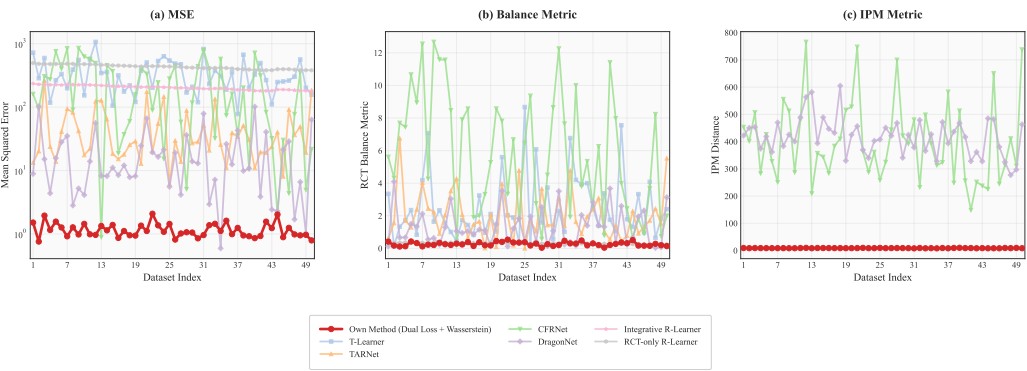

*Figure 2.* Robustness under Severe Conditional Non-overlap

Methods that rely on rigid structural assumptions (e.g., Experimental Grounding (Kallus et al., 2018)) exhibit substantial performance degradation under severe non-overlap and are therefore omitted from this comparison.

As shown in Figure 2, the proposed method consistently achieves the lowest MSE across all datasets while simultaneously maintaining feasibility, as characterized by Theorem 3.7. In contrast, baseline methods either suffer from large estimation error or fail to preserve feasibility under severe support mismatch. Notably, classical distribution alignment approaches such as CFRNet reduce the IPM distance but do so at the cost of significantly degraded predictive accuracy, indicating that marginal distribution alignment alone is insufficient in the presence of conditional non-overlap.

We also conducted a sensitivity analysis (see Appendix H) on several key hyperparameters of the proposed framework, including the dual update coefficient, the dual variable weight $\nu$, the augmentation parameter $\rho$, and the Wasserstein regularization weight. Under different degrees of overlap, within a given interval, it can robustly satisfy the moment condition and achieve a good level in the AUUC index. At the same time, it can be seen from the training loss graph that the training is stable.

## 5. Experiments

We evaluate the proposed joint training approach on large-scale real-world data from a ride-hailing platform. In this setting, observational data are abundant, with millions of samples generated daily, whereas randomized controlled trial (RCT) data are costly and limited. The resulting data exhibit high-dimensional covariates (hundreds of features) and pronounced *joint distribution mismatch* between observational and randomized data. Importantly, this mismatch differs from Marginal non-overlap (Definition 2.1): all treatment arms observed in the RCT also appear in the observational data, but their joint distributions with covariates differ substantially.

*Table 2.* Performance comparison across datasets.

| Method | Dataset 1 | | Dataset 2 | | Dataset 3 | | Dataset 4 | |
|---|---|---|---|---|---|---|---|---|
| | Qini | MAPE | Qini | MAPE | Qini | MAPE | Qini | MAPE |
| Baseline | 0.64079 | 0.15002 | 0.64523 | 0.14563 | 0.63202 | 0.09419 | 0.65411 | 0.09166 |
| OBS-only | 0.60668 | 0.19851 | 0.61962 | 0.18608 | 0.60597 | 0.17534 | 0.63951 | 0.15832 |
| RCT-only | 0.64341 | 0.08721 | 0.65172 | 0.05130 | 0.65313 | 0.06183 | 0.65667 | 0.14068 |
| Own-method | 0.65073 | 0.07772 | 0.65439 | 0.02883 | 0.66075 | 0.06166 | 0.66145 | 0.04827 |
| Own-method-rep | 0.64842 | 0.09018 | 0.64756 | 0.04494 | 0.60599 | 0.11419 | 0.65322 | 0.06207 |
| Own-method-cons | 0.63636 | 0.06639 | 0.65141 | 0.09744 | 0.64959 | 0.06438 | 0.65451 | 0.06024 |

*Note*: All data have been desensitized.

### 5.1. Experimental Setup and Results

We randomly sample four observational datasets collected over four different months and combine each with samples from a shared RCT dataset spanning six months, thereby inducing additional covariate shift and increasing the difficulty of the estimation problem. Each dataset contains $n = 10,000,000$ samples, closely reflecting real-world deployment conditions.

Results are reported in Table 2. We report commercially relevant metrics, including the Qini coefficient and MAPE. Our primary baseline is the ride-hailing platform's current production model (details omitted for confidentiality); additional internal baselines are omitted as they have proven worser than the latest by the platform. We also include RCT-only and OBS-only variants trained solely on randomized and observational data, respectively.

To evaluate the role of each component, we also perform an ablation study by removing individual modules of the proposed framework in Table 2. Specifically, REP uses representation learning without constrained estimation, while CONS enforces the constrained program without representation learning. Table 2 shows that the proposed joint framework consistently outperforms OBS-only, RCT-only, and production baselines, with larger gains on Datasets 3–4, where mixed (RCT+OBS) training achieves performance comparable to substantially larger RCT-only training, suggesting improved efficiency with reduced experimental cost.

## Acknowledgment

Zhiheng Zhang is supported by "the Fundamental Research Funds for the Central Universities" (Grant No. 2025110602) of Shanghai University of Finance and Economics, and Independent Research Project (Grant No. 2026110081) funded by the School of Statistics and Data Science. H. Li is supported by the Beijing Major Science and Technology Project (No. Z251100008425006) and the Beijing Natural Science Foundation (No. L257007). C. Fang is supported by the National Natural Science Foundation of China (No.s 92470117, 62376008). This work was supported by the Shanghai Engineering Research Center of Finance Intelligence under Grant No. 19DZ2254600, the State Key Laboratory of General Artificial Intelligence, and the CCF-DiDi GAIA Collaborative Research Fund under Grant No. CCF-DiDi GAIA 202517, and was conducted as part of the industry-academia collaboration among DiDi, Peking University, and Shanghai University of Finance and Economics. We also appreciate the guidance provided by the engineers in Didichuxing Co. Ltd. We are sincerely grateful to Chengchun Shi for the important inspiration provided for this work.

## Impact Statement

This paper presents work whose goal is to advance the field of machine learning. There are many potential societal consequences of our work, none of which we feel must be specifically highlighted here.

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

## A. Literature Review

A substantial literature investigates how to combine randomized controlled trials (RCTs) with observational (OBS) data in order to improve the estimation of causal effects, motivated by the complementary strengths of internal validity from experiments and statistical efficiency from large-scale logs. Broadly, existing approaches treat RCT data as a source of causal identification and OBS data as a vehicle for variance reduction, covariate enrichment, or limited extrapolation. Despite important progress, these methods typically rely—either explicitly or implicitly—on overlap or transportability conditions that are violated in many modern platform settings, particularly when individual-level effects under structured treatments are of primary interest. This work is also motivated by our lab's previous exploration (Zhang & Su, 2024; Zhang, 2022; Zhang et al., 2023; Zhang, 2024; Su & Zhang et al, 2025; Wang et al., 2025; Zhang et al., 2025; Zhang & Wang, 2025; Li et al., 2026; Su et al., 2023; 2024; 2026; Lu et al., 2026; Zhang, 2026).

**Experimental grounding and structural decompositions.** Early integrative frameworks, such as experimental grounding (Kallus et al., 2018), leverage RCTs to correct bias in models trained on observational data by parameterizing hidden confounding and estimating the correction using experimental samples. This approach weakens the ignorability requirement and enables partial extrapolation beyond experimental support. However, its validity fundamentally depends on low-complexity structural assumptions on the confounding function. When overlap between RCT and OBS covariate distributions is severely limited, the correction term becomes weakly identified, rendering the procedure highly sensitive to model misspecification and offering no protection against structural absence of support. More recent semiparametric formulations (Yang et al., 2025) formalize RCT–OBS integration by decomposing observed effects into a heterogeneous treatment effect (HTE) and a confounding component, establishing identification and efficiency gains under transportability and parametric modeling assumptions. While theoretically elegant, these guarantees are derived for finite-dimensional projections and hinge on correct specification of the structural components; under strong non-overlap, estimation necessarily relies on extrapolation, which limits robustness and portability to high-dimensional settings.

**Adaptive fusion and orthogonal learning.** A complementary line of work seeks to safeguard against observational bias through adaptive combination schemes. The elastic pre-test framework (Yang et al., 2023) dynamically downweights or discards observational data when comparability tests fail, thereby protecting against severe bias. Yet, when violations are detected, the resulting estimand effectively collapses to an RCT-only target, sacrificing individual-level resolution and providing no mechanism to recover effects outside experimental support. Similarly, integrative R-learner approaches (Wu & Yang, 2022) employ Neyman-orthogonal losses to jointly estimate heterogeneous treatment effects and confounding functions using flexible learners, achieving efficiency gains relative to RCT-only estimation. Nonetheless, their theoretical guarantees remain confined to the covariate support of the RCT; observational data primarily reduce variance but do not resolve fundamental support mismatches, and identification of individual treatment effects beyond the experimental region remains out of reach.

**Limitations under severe non-overlap.** Across these paradigms, RCT data are consistently treated as the gold standard for identification, while OBS data are incorporated through weighted losses, orthogonalization, or pre-testing. Despite methodological differences, these approaches share a common limitation: they do not fundamentally address settings in which overlap between RCT and OBS populations is *structurally* violated. In such regimes, covariate–treatment combinations may be entirely absent from observational logs due to policy or design constraints, and no amount of reweighting, tuning, or sample-size increase can recover the missing support. As a result, existing methods either rely on extrapolation driven by structural assumptions or retreat to coarse estimands defined solely on experimental support, limiting their applicability to large-scale decision systems where individual-level personalization is essential.

## B. Proof

**Setup.** Let $\mathcal{M}$ be a (possibly over-parameterized) class of measurable predictors $m : \mathcal{X} \times \mathcal{T} \to \mathbb{R}$. Define the observational and randomized risks as

$$\mathcal{R}_o(m) := \mathbb{E}_{P_o}\big[(Y - m(X,T))^2\big], \qquad \mathcal{R}_r(m) := \mathbb{E}_{P_r}\big[(Y - m(X,T))^2\big].$$

Let the identifying (randomization-induced) moment map be

$$g(m) := \mathbb{E}_{P_r}\big[\psi(Y,T,X;m)\big] \in \mathbb{R}^K, \qquad \mathcal{F} := \{m \in \mathcal{M} : g(m) = 0\}.$$

The constrained estimator is defined as

$$m^\star \in \arg\min_{m \in \mathcal{M}} \mathcal{R}_o(m) \quad \text{s.t.} \quad g(m) = 0, \qquad \mathcal{R}_o^\star := \mathcal{R}_o(m^\star) = \inf_{m \in \mathcal{F}} \mathcal{R}_o(m). \tag{20}$$

The weighted-loss fusion estimator is, for $\alpha \geq 0$,

$$m_\alpha \in \arg\min_{m \in \mathcal{M}} J(m, \alpha) := \mathcal{R}_o(m) + \alpha \, \mathcal{R}_r(m). \tag{21}$$

**Assumption B.1** (Strong convexity and smoothness of the weighted objective). Fix $\alpha_{\max} > 0$. For every $\alpha \in [0, \alpha_{\max}]$, the weighted objective

$$J(m, \alpha) = \mathcal{R}_o(m) + \alpha \, \mathcal{R}_r(m)$$

is $\mu$-strongly convex over $\mathcal{M}$ and twice continuously differentiable. Moreover, $\|\nabla \mathcal{R}_r(m)\| \leq M_r$ for all $m \in \mathcal{M}$.

**Assumption B.2** (Lipschitz continuity of the moment map). The identifying moment map $g : \mathcal{M} \to \mathbb{R}^K$ defined by $g(m) = \mathbb{E}_{P_r}[\psi(Y, T, X; m)]$ is $L_g$-Lipschitz:

$$\|g(m) - g(m')\| \leq L_g \, \|m - m'\|_{L_2(P_r)} \qquad \forall m, m' \in \mathcal{M}.$$

**Assumption B.3** (Error bound and constraint regularity). There exist constants $\mu_o > 0$ and $L_o > 0$ such that for all $m \in \mathcal{M}$,

$$\mathcal{R}_o(m) - \mathcal{R}_o^\star \geq \mu_o \, \text{dist}(m, \mathcal{F})^2, \tag{22}$$
$$\|g(m)\| \geq L_o \, \text{dist}(m, \mathcal{F}), \tag{23}$$

where $\mathcal{F} := \{m \in \mathcal{M} : g(m) = 0\}$ and $\text{dist}(m, \mathcal{F}) := \inf_{m' \in \mathcal{F}} \|m - m'\|_{L_2(P_r)}$.

**Assumption B.4** (Moment regularity under representation perturbations). Let $\mathcal{D}$ be the discriminator class defining $\text{IPM}_{\mathcal{D}}$. There exists $L_\psi > 0$ such that for any representations $\phi_1, \phi_2$ and any $m \in \mathcal{M}$,

$$\left\| \mathbb{E}_{P_r}[\psi(Y, T, \phi_1(X); m)] - \mathbb{E}_{P_r}[\psi(Y, T, \phi_2(X); m)] \right\| \leq L_\psi \, \text{IPM}_{\mathcal{D}}(P_r(\phi_1(X), T), \, P_r(\phi_2(X), T)) .$$

**Assumption B.5** (Moment misspecification in the original space. ). There exist $k \in \{1, \ldots, K\}$ and $\delta > 0$ such that for all $m \in \mathcal{M}$,

$$\left| \mathbb{E}_{P_r}[Y - m(X, T) \mid (X, T) \in S] \right| \geq \delta.$$

**Assumption B.6** (Non-degenerate treatment assignment). For $(X, T) \in S$,

$$\underline{p} \leq P_r(T = k \mid X) \leq \bar{p} \quad \text{for some } 0 < \underline{p} < \bar{p} < 1.$$

**Assumption B.7** (Overlap-to-feasibility control). There exists $c_{\text{ov}} > 0$ such that for every representation $\phi$,

$$\inf_m \|g(m, \phi)\| \leq c_{\text{ov}} \, \varepsilon_{\text{ov}}(\phi),$$

where $\varepsilon_{\text{ov}}(\phi)$ denotes the population overlap (IPM) discrepancy.

**Assumption B.8** (Statistical estimation error). Let $\widehat{\mathcal{R}}_o$ and $\widehat{g}$ be empirical estimates based on $n_o$ observational and $n_r$ randomized samples. There exists $\text{Stat}(n_o, n_r)$ such that with probability at least $1 - \delta$,

$$\sup_{m, \phi} \left| \widehat{\mathcal{R}}_o(m, \phi) - \mathcal{R}_o(m, \phi) \right| + \sup_{m, \phi} \left\| \widehat{g}(m, \phi) - g(m, \phi) \right\| \leq \text{Stat}(n_o, n_r).$$

**Proposition B.9** (Lipschitz solution path for weighted fusion). *Under Assumption B.1, for each $\alpha \in [0, \alpha_{\max}]$ the weighted-loss minimizer $m_\alpha$ of $J(m, \alpha) = \mathcal{R}_o(m) + \alpha \mathcal{R}_r(m)$ is unique, and the solution path $\alpha \mapsto m_\alpha$ is differentiable with*

$$\left\| \frac{dm_\alpha}{d\alpha} \right\| \leq \frac{M_r}{\mu}, \qquad \forall \alpha \in [0, \alpha_{\max}].$$

*Consequently,*

$$\|m_\alpha - m_0\| \leq \frac{M_r}{\mu} \alpha. \tag{24}$$

*Proof.* The first-order optimality condition is $\nabla_m J(m_\alpha, \alpha) = 0$. Differentiating with respect to $\alpha$ and applying the implicit function theorem yields

$$\nabla^2_{mm} J(m_\alpha, \alpha) \frac{dm_\alpha}{d\alpha} + \nabla \mathcal{R}_r(m_\alpha) = 0,$$

so that $\frac{dm_\alpha}{d\alpha} = -(\nabla^2_{mm} J(m_\alpha, \alpha))^{-1} \nabla \mathcal{R}_r(m_\alpha)$. By $\mu$-strong convexity, $\|(\nabla^2_{mm} J)^{-1}\| \leq 1/\mu$, and by Assumption B.1, $\|\nabla \mathcal{R}_r(m_\alpha)\| \leq M_r$, which gives the derivative bound. Integrating over $\alpha$ yields (24). $\qquad \square$

**Theorem B.10** (Irreducible overlap under marginal structural non-overlap). *Suppose marginal structural non-overlap holds in the sense of Definition 2.1, i.e., there exists a measurable $A \subseteq \mathcal{T}$ such that $P_r(T \in A) > 0$ and $P_o(T \in A) = 0$. Assume the discriminator class $\mathcal{D}$ contains functions depending only on $T$, in particular the indicator $f_A(z, t) = \mathbf{1}\{t \in A\}$.*

*Then, for any representation $\phi : \mathcal{X} \to \mathcal{Z}$, the overlap discrepancy*

$$\varepsilon_{ov}(\phi) = \text{IPM}_{\mathcal{D}}\big(P_o(\phi(X), T), P_r(\phi(X), T)\big)$$

*satisfies*

$$\varepsilon_{ov}(\phi) \geq P_r(T \in A) > 0.$$

*Consequently,*

$$\inf_{\phi \in \Phi} \varepsilon_{ov}(\phi) \geq P_r(T \in A),$$

*and overlap recovery is impossible under marginal structural non-overlap.*

*Proof.* By definition of the IPM,

$$\varepsilon_{ov}(\phi) = \sup_{f \in \mathcal{D}} \Big| \mathbb{E}_{P_o}[f(\phi(X), T)] - \mathbb{E}_{P_r}[f(\phi(X), T)] \Big|.$$

Since $f_A(z, t) = \mathbf{1}\{t \in A\} \in \mathcal{D}$, we obtain

$$\varepsilon_{ov}(\phi) \geq \big| P_o(T \in A) - P_r(T \in A) \big|.$$

Under marginal structural non-overlap, $P_o(T \in A) = 0$ and $P_r(T \in A) > 0$, which yields the claim. Taking the infimum over $\phi$ preserves the bound. $\qquad \square$

Throughout, we focus on conditional structural non-overlap (Definition 2.2), under which overlap violations may be repaired through representation learning. Marginal structural non-overlap (Definition 2.1) is fundamentally irreducible, as formalized in Theorem B.10.

**Proof of Theorem 3.6**

*Proof.* Fix any $m \in \mathcal{M}$. By Assumption B.5, there exist $k \in \{1, \ldots, K\}$ and $\delta > 0$ such that

$$\big| \mathbb{E}_{P_r}[Y - m(X, T) \mid (X, T) \in S] \big| \geq \delta.$$

Let $\psi_k$ denote the $k$-th coordinate of the moment map, i.e.,

$$\psi_k(Y, T, X; m) = (D_k - p_k)\big(Y - m(X, T)\big), \quad D_k := \mathbf{1}\{T = k\}, \quad p_k := P_r(T = k).$$

Define $U := (D_k - p_k)\big(Y - m(X, T)\big) \mathbf{1}_S$. Then

$$\mathbb{E}_{P_r}[\psi_k(Y, T, X; m)] = \mathbb{E}_{P_r}[U] + \mathbb{E}_{P_r}\big[(D_k - p_k)(Y - m(X, T)) \mathbf{1}_{S^c}\big],$$

and therefore

$$\big| \mathbb{E}_{P_r}[\psi_k(Y, T, X; m)] \big| \geq \big| \mathbb{E}_{P_r}[U] \big| - \big| \mathbb{E}_{P_r}\big[(D_k - p_k)(Y - m(X, T)) \mathbf{1}_{S^c}\big] \big|.$$

In particular, it suffices to lower bound $\big| \mathbb{E}_{P_r}[U] \big|$.

By the law of total expectation,

$$\mathbb{E}_{P_r}[U] = P_r(S) \cdot \mathbb{E}_{P_r}\Big[(D_k - p_k)(Y - m(X,T)) \,\Big|\, S\Big].$$

On $S$, Assumption B.6 yields $\underline{p} \leq P_r(T = k \mid X) \leq \bar{p}$ and hence

$$\mathbb{E}_{P_r}[D_k - p_k \mid S] = \mathbb{E}_{P_r}\big[P_r(T = k \mid X) - p_k \mid S\big],$$

which has a fixed sign on $S$ after choosing the coordinate $k$ in Assumption B.5 (equivalently, flip the sign of $\psi_k$ if needed). Moreover, under the same non-degeneracy bounds,

$$\mathbb{E}_{P_r}\big[|D_k - p_k| \mid S\big] \;\geq\; \underline{p}(1 - \bar{p}).$$

Combining with Assumption B.5 (fixing the sign so that the conditional mean is $\geq \delta$), we obtain

$$\left|\mathbb{E}_{P_r}\Big[(D_k - p_k)(Y - m(X,T)) \,\Big|\, S\Big]\right| \;\geq\; \underline{p}(1 - \bar{p}) \cdot \left|\mathbb{E}_{P_r}\big[Y - m(X,T) \mid S\big]\right| \;\geq\; \underline{p}(1 - \bar{p})\,\delta.$$

Therefore,

$$\left|\mathbb{E}_{P_r}[U]\right| \;\geq\; P_r(S)\,\underline{p}(1 - \bar{p})\,\delta.$$

Since the above bound holds for all $m \in \mathcal{M}$, we conclude that the feasibility gap in the original space is strictly positive:

$$\mathcal{F}_X(\mathcal{M}) \;:=\; \inf_{m \in \mathcal{M}} \big\|\mathbb{E}_{P_r}[\psi(Y,T,X;m)]\big\| \;\geq\; P_r(S)\,\underline{p}(1 - \bar{p})\,\delta \;=:\; c_0 \;>\; 0.$$

$\square$

**Corollary B.11** (Moment residual decreases at most linearly along the weighted path). *Under Assumptions B.1–B.2 and Proposition B.9, for any $\alpha \in [0, \alpha_{\max}]$,*

$$\|g(m_\alpha)\| \;\geq\; \|g(m_0)\| - \frac{L_g M_r}{\mu}\,\alpha. \tag{25}$$

*In particular, achieving $\|g(m_\alpha)\| \approx 0$ requires $\alpha \gtrsim \mu\|g(m_0)\|/(L_g M_r)$, i.e., placing dominant weight on the randomized objective.*

*Proof.* By Assumption B.2, the moment map is $L_g$-Lipschitz with respect to the $L_2(P_r)$-norm:

$$\|g(m_\alpha) - g(m_0)\| \;\leq\; L_g\,\|m_\alpha - m_0\|.$$

Applying the reverse triangle inequality yields

$$\|g(m_\alpha)\| \;\geq\; \|g(m_0)\| - \|g(m_\alpha) - g(m_0)\| \;\geq\; \|g(m_0)\| - L_g\,\|m_\alpha - m_0\|.$$

Finally, Proposition B.9 bounds the solution path:

$$\|m_\alpha - m_0\| \;\leq\; \frac{M_r}{\mu}\,\alpha, \qquad \forall\,\alpha \in [0, \alpha_{\max}],$$

which implies (25). The final statement follows by rearranging (25) to make the right-hand side nonpositive, i.e., $\alpha \geq \mu\|g(m_0)\|/(L_g M_r)$ up to constant factors. $\square$

**Proof of Corollary 3.7.**

*Proof.* We prove the two claims in turn.

**(1) Strict feasibility improvement.** By the overlap-to-feasibility control (Assumption B.7),

$$\mathcal{F}_{\phi^\star}(\mathcal{M}) \;\leq\; c_1\,\varepsilon_{ov}(\phi^\star).$$

If $\varepsilon_{ov}(\phi^\star) < c_0/c_1$, then

$$\mathcal{F}_{\phi^\star}(\mathcal{M}) \;\leq\; c_1\varepsilon_{ov}(\phi^\star) \;<\; c_0 \;\leq\; \mathcal{F}_X(\mathcal{M}),$$

where the last inequality uses the raw-space obstruction $\mathcal{F}_X(\mathcal{M}) \geq c_0$. This establishes strict feasibility improvement.

**(2) Excess-risk lower bound via quadratic growth and Lipschitz moments.** Fix $\phi^\star$ and assume $\mathcal{F}_{\phi^\star} \neq \emptyset$ so that $\mathcal{R}_{o,\phi^\star}^{\mathrm{feas}} := \inf_{m \in \mathcal{F}_{\phi^\star}} \mathcal{R}_o(m)$ is well-defined. By quadratic growth(Assumption B.3) (with $\phi = \phi^\star$), for all $m \in \mathcal{M}_{\phi^\star}$,

$$\mathcal{R}_o(m) - \mathcal{R}_{o,\phi^\star}^{\mathrm{feas}} \;\geq\; \mu_o \, \mathrm{dist}(m, \mathcal{F}_{\phi^\star})^2. \tag{26}$$

By the Lipschitz moment condition (B.2) (with $\phi = \phi^\star$), for all $m \in \mathcal{M}_{\phi^\star}$,

$$\|g_{\phi^\star}(m)\| \;\leq\; L_g \, \mathrm{dist}(m, \mathcal{F}_{\phi^\star}) \qquad \Longrightarrow \qquad \mathrm{dist}(m, \mathcal{F}_{\phi^\star}) \;\geq\; \frac{\|g_{\phi^\star}(m)\|}{L_g}. \tag{27}$$

Combining (26) and (27) yields, for all $m \in \mathcal{M}_{\phi^\star}$,

$$\mathcal{R}_o(m) - \mathcal{R}_{o,\phi^\star}^{\mathrm{feas}} \;\geq\; \frac{\mu_o}{L_g^2} \, \|g_{\phi^\star}(m)\|^2.$$

Taking the infimum over $m \in \mathcal{M}_{\phi^\star}$ and using the definition $\mathcal{F}_{\phi^\star}(\mathcal{M}) = \inf_{m \in \mathcal{M}_{\phi^\star}} \|g_{\phi^\star}(m)\|$ gives

$$\inf_{m \in \mathcal{M}_{\phi^\star}} \left\{ \mathcal{R}_o(m) - \mathcal{R}_{o,\phi^\star}^{\mathrm{feas}} \right\} \;\geq\; \frac{\mu_o}{L_g^2} \left( \mathcal{F}_{\phi^\star}(\mathcal{M}) \right)^2,$$

which is the first inequality in (17). The second inequality in (17) follows from Assumption B.7: $\mathcal{F}_{\phi^\star}(\mathcal{M}) \leq c_1 \varepsilon_{ov}(\phi^\star)$. This completes the proof. $\qquad\square$

*Remark* B.12. Corollary 3.7 shows that whenever $\varepsilon^\star < c_0/c_1$, such a solution strictly improves upon any estimator defined in the original covariate space. Hence, introducing the representation $\phi$ is both necessary (to overcome structural infeasibility) and sufficient (to achieve strict risk improvement) under the stated assumptions.

**Theorem B.13** (Minimax lower bound for feasibility recovery). *Fix any discriminator class $\mathcal{D}$ and define the overlap discrepancy*

$$\varepsilon_{ov}(\phi) := \mathrm{IPM}_{\mathcal{D}}\big( P_o(\phi(X), T), \; P_r(\phi(X), T) \big).$$

*Assume $\mathcal{D}$ contains indicators of measurable sets on $\mathcal{Z} \times \mathcal{T}$ (so that $\mathrm{IPM}_{\mathcal{D}}$ dominates $\mathrm{TV}$ on $(Z,T)$), and $|Y| \leq 1$.*

*Then for any $\varepsilon \in (0,1)$, there exist distributions $(P_o, P_r)$ and a model class $\mathcal{M}$ such that:*

1. *There exists $\phi$ with $\varepsilon_{ov}(\phi) = \varepsilon$.*

2. *For every $\phi$ satisfying $\varepsilon_{ov}(\phi) \leq \varepsilon$, the feasibility gap*

$$\mathcal{F}_\phi(\mathcal{M}) := \inf_{m \in \mathcal{M}_\phi} \left\| \mathbb{E}_{P_r}\big[ \psi(Y, T, \phi(X); m) \big] \right\|$$

*obeys $\mathcal{F}_\phi(\mathcal{M}) \geq c\varepsilon$ for a universal constant $c > 0$.*

*Consequently, the $\mathcal{O}(\varepsilon_{ov})$ dependence is minimax-tight up to constants.*

**Proof of Theorem B.13.**

*Proof.* We give an explicit finite construction. Let $\mathcal{T} = \{0,1\}$ and let the latent variable $Z \in \{a,b\}$. Take a representation class that can reveal $Z$, and consider $\phi(X) = Z$.

**Step 1: Construct $(P_o, P_r)$ with $\varepsilon_{ov}(\phi) = \varepsilon$.** Define the randomized joint distribution on $(Z,T)$ by

$$P_r(Z = a) = P_r(Z = b) = \tfrac{1}{2}, \qquad P_r(T = 1 \mid Z) = \tfrac{1}{2},$$

so $P_r(Z = z, T = t) = \tfrac{1}{4}$ for all $(z,t)$. Define the observational joint distribution by shifting $\varepsilon$ mass within the treated arm:

$$P_o(Z = a, T = 1) = \tfrac{1}{4} - \tfrac{\varepsilon}{2}, \qquad P_o(Z = b, T = 1) = \tfrac{1}{4} + \tfrac{\varepsilon}{2},$$

and set $P_o(Z,T) = P_r(Z,T)$ on the remaining three cells. Then $\mathrm{TV}(P_o(Z,T), P_r(Z,T)) = \varepsilon$. Since $\mathcal{D}$ contains indicators, $\mathrm{IPM}_{\mathcal{D}} \geq \mathrm{TV}$, hence

$$\varepsilon_{ov}(\phi) = \mathrm{IPM}_{\mathcal{D}}\big(P_o(Z,T), P_r(Z,T)\big) \geq \varepsilon.$$

By scaling the shift by a constant factor (absorbed into $c$ below), we may assume $\varepsilon_{ov}(\phi) = \varepsilon$.

**Step 2: Choose outcomes and a misspecified model class.** Let $Y \in \{-1, +1\}$ with

$$\mathbb{E}_{P_r}[Y \mid Z = a, T = 1] = +1, \qquad \mathbb{E}_{P_r}[Y \mid Z = b, T = 1] = -1, \qquad \mathbb{E}_{P_r}[Y \mid T = 0, Z] = 0,$$

so $|Y| \leq 1$. Consider the restricted model class $\mathcal{M}$ that cannot use $Z$ under treatment:

$$m(z,t) = \begin{cases} 0, & t = 0, \\ \alpha, & t = 1, \end{cases} \qquad \alpha \in [-1, 1].$$

Under the induced class $\mathcal{M}_\phi$ (with $\phi(X) = Z$), predictors remain constant on $(Z, T = 1)$.

**Step 3: A linear functional lower bounds feasibility by a $(Z,T)$-imbalance.** We consider a single-coordinate moment map of the form

$$\psi(Y, T, Z; m) := \mathbf{1}\{T = 1\}\big(Y - m(Z, 1)\big),$$

so that $g(m, \phi) = \mathbb{E}_{P_r}[\psi(Y, T, \phi(X); m)]$ reduces to the treated-arm mean residual. (Any IPW-style moment family that contains such a coordinate yields the same conclusion up to constants.)

For any $m \in \mathcal{M}_\phi$ with treated prediction $\alpha$,

$$g(m, \phi) = \mathbb{E}_{P_r}\big[\mathbf{1}\{T = 1\}(Y - \alpha)\big] = \sum_{z \in \{a, b\}} P_r(Z = z, T = 1)\big(\mathbb{E}[Y \mid z, 1] - \alpha\big).$$

Using $\mathbb{E}[Y \mid a, 1] = +1$ and $\mathbb{E}[Y \mid b, 1] = -1$, we obtain

$$g(m, \phi) = P_r(a, 1)(1 - \alpha) + P_r(b, 1)(-1 - \alpha) = \big(P_r(a, 1) - P_r(b, 1)\big) - \alpha\big(P_r(a, 1) + P_r(b, 1)\big).$$

Since $P_r(a, 1) + P_r(b, 1) = P_r(T = 1) = \frac{1}{2}$, optimizing over $\alpha \in [-1, 1]$ yields

$$\inf_{\alpha \in [-1, 1]} |g(m, \phi)| \geq \big|P_r(a, 1) - P_r(b, 1)\big| - \tfrac{1}{2}.$$

To make the lower bound proportional to $\varepsilon$, we now choose $P_r$ with an $\varepsilon$-imbalance in the treated arm:

$$P_r(Z = a, T = 1) = \tfrac{1}{4} + \tfrac{\varepsilon}{2}, \qquad P_r(Z = b, T = 1) = \tfrac{1}{4} - \tfrac{\varepsilon}{2},$$

while keeping $P_r(T = 1) = \frac{1}{2}$. Then $|P_r(a, 1) - P_r(b, 1)| = \varepsilon$ and the minimizer over $\alpha$ cannot cancel this imbalance because $\alpha$ is constant across $Z$ under $T = 1$, hence

$$\inf_{m \in \mathcal{M}_\phi} \big|g(m, \phi)\big| \geq c_0 \varepsilon$$

for an absolute constant $c_0 > 0$ (here one may take $c_0 = 1$ for the above coordinate).

**Step 4: Any $\tilde{\phi}$ with $\varepsilon_{ov}(\tilde{\phi}) \leq \varepsilon$ cannot remove the $\Theta(\varepsilon)$ imbalance.** Because $\mathrm{IPM}_{\mathcal{D}}$ dominates TV on $(\tilde{Z}, T)$ with $\tilde{Z} = \tilde{\phi}(X)$, the condition $\varepsilon_{ov}(\tilde{\phi}) \leq \varepsilon$ implies $\mathrm{TV}\big(P_o(\tilde{Z}, T), P_r(\tilde{Z}, T)\big) \leq \varepsilon$. In particular, for the measurable set $A := \{\tilde{Z}$ corresponds to $Z = b, \ T = 1\}$ (which is representable since $\mathcal{D}$ contains indicators), the mass discrepancy on $A$ is at most $\varepsilon$. Thus any representation that reduces the overlap discrepancy below $\varepsilon$ can only reduce the treated-arm imbalance (and hence the moment residual above) by at most a constant factor, yielding

$$\mathcal{F}_{\tilde{\phi}}(\mathcal{M}) = \inf_{m \in \mathcal{M}_{\tilde{\phi}}} \big\|\mathbb{E}_{P_r}[\psi(Y, T, \tilde{\phi}(X); m)]\big\| \geq c\varepsilon$$

for a universal constant $c > 0$. This proves the minimax lower bound and the claimed tightness. $\square$

**Proposition B.14** (Feasibility guarantee of joint primal–dual training). *Consider the population constrained program*

$$\min_{\theta,\omega} \ R_o(\theta,\omega) \quad s.t. \quad g(\theta,\omega) = 0, \tag{28}$$

*where $g(\theta,\omega) := \mathbb{E}_{P_r}[\psi(Y,T,\omega(X); m_\theta)] \in \mathbb{R}^K$. For $\Lambda > 0$, define the restricted Lagrangian*

$$\mathcal{L}_\Lambda(\theta,\omega,\lambda) := R_o(\theta,\omega) + \lambda^\top g(\theta,\omega), \qquad \lambda \in \mathbb{B}_\Lambda := \{\lambda \in \mathbb{R}^K : \ \|\lambda\|_2 \le \Lambda\}.$$

*Assume:*

1. *$\{(\theta,\omega) : g(\theta,\omega) = 0\} \neq \emptyset$;*

2. *For each fixed $\omega$, $R_o(\theta,\omega)$ is $C^1$ and $\mu$-strongly convex in $\theta$;*

3. *$g(\theta,\omega)$ is continuous and $L_g$-Lipschitz in $\theta$.*

*Then any saddle point $(\theta^\star, \omega^\star, \lambda^\star)$ of*

$$\min_{\theta,\omega} \ \max_{\lambda \in \mathbb{B}_\Lambda} \ \mathcal{L}_\Lambda(\theta,\omega,\lambda)$$

*is feasible, i.e., $g(\theta^\star, \omega^\star) = 0$.*

*Moreover, if $(\hat{\theta}, \hat{\omega}, \hat{\lambda})$ is an $\varepsilon_{\mathrm{opt}}$-approximate saddle point:*

$$\max_{\lambda \in \mathbb{B}_\Lambda} \mathcal{L}_\Lambda(\hat{\theta}, \hat{\omega}, \lambda) - \min_{\theta,\omega} \mathcal{L}_\Lambda(\theta,\omega,\hat{\lambda}) \le \varepsilon_{\mathrm{opt}},$$

*then*

$$\|g(\hat{\theta}, \hat{\omega})\|_2 \le \varepsilon_{\mathrm{opt}}/\Lambda. \tag{29}$$

*Proof.* **(a) Exact saddle point implies feasibility.** Let $(\theta^\star, \omega^\star, \lambda^\star)$ be a saddle point. Then for all $(\theta, \omega)$ and all $\lambda \in \mathbb{B}_\Lambda$,

$$\mathcal{L}_\Lambda(\theta^\star, \omega^\star, \lambda) \le \mathcal{L}_\Lambda(\theta^\star, \omega^\star, \lambda^\star) \le \mathcal{L}_\Lambda(\theta, \omega, \lambda^\star). \tag{30}$$

Pick any feasible $(\tilde{\theta}, \tilde{\omega})$ with $g(\tilde{\theta}, \tilde{\omega}) = 0$ (exists by (i)). The right inequality in (30) yields

$$\mathcal{L}_\Lambda(\theta^\star, \omega^\star, \lambda^\star) \le \mathcal{L}_\Lambda(\tilde{\theta}, \tilde{\omega}, \lambda^\star) = R_o(\tilde{\theta}, \tilde{\omega}).$$

On the other hand, the left inequality in (30) implies

$$\max_{\lambda \in \mathbb{B}_\Lambda} \mathcal{L}_\Lambda(\theta^\star, \omega^\star, \lambda) \le \mathcal{L}_\Lambda(\theta^\star, \omega^\star, \lambda^\star).$$

Since $\max_{\|\lambda\|_2 \le \Lambda} \lambda^\top g = \Lambda \|g\|_2$, we have

$$R_o(\theta^\star, \omega^\star) + \Lambda \|g(\theta^\star, \omega^\star)\|_2 \le \mathcal{L}_\Lambda(\theta^\star, \omega^\star, \lambda^\star) \le R_o(\tilde{\theta}, \tilde{\omega}).$$

If $g(\theta^\star, \omega^\star) \neq 0$, the left-hand side is strictly larger than $R_o(\theta^\star, \omega^\star)$, which contradicts optimality of the feasible primal solution for fixed $\Lambda > 0$. Hence $g(\theta^\star, \omega^\star) = 0$.

**(b)Approximate saddle point controls feasibility residual.** Let $(\hat{\theta}, \hat{\omega}, \hat{\lambda})$ satisfy the $\varepsilon_{\mathrm{opt}}$ saddle gap condition. Using again $\max_{\|\lambda\|_2 \le \Lambda} \lambda^\top g = \Lambda \|g\|_2$,

$$\max_{\lambda \in \mathbb{B}_\Lambda} \mathcal{L}_\Lambda(\hat{\theta}, \hat{\omega}, \lambda) = R_o(\hat{\theta}, \hat{\omega}) + \Lambda \|g(\hat{\theta}, \hat{\omega})\|_2.$$

For any feasible $(\tilde{\theta}, \tilde{\omega})$,

$$\min_{\theta,\omega} \mathcal{L}_\Lambda(\theta,\omega,\hat{\lambda}) \le \mathcal{L}_\Lambda(\tilde{\theta}, \tilde{\omega}, \hat{\lambda}) = R_o(\tilde{\theta}, \tilde{\omega}).$$

Therefore,

$$R_o(\hat{\theta}, \hat{\omega}) + \Lambda \|g(\hat{\theta}, \hat{\omega})\|_2 - R_o(\tilde{\theta}, \tilde{\omega}) \le \varepsilon_{\mathrm{opt}}.$$

Dropping $R_o(\hat{\theta}, \hat{\omega}) - R_o(\tilde{\theta}, \tilde{\omega}) \ge -\infty$ but keeping the inequality valid, we obtain $\Lambda \|g(\hat{\theta}, \hat{\omega})\|_2 \le \varepsilon_{\mathrm{opt}}$, proving (29). $\quad\square$

**Theorem B.15** (Feasibility–information trade-off (incremental form)). *Let $\mathcal{M}$ be a model class of measurable predictors $m : \mathcal{X} \times \mathcal{T} \to \mathbb{R}$. For any representation $\phi : \mathcal{X} \to \mathcal{Z}$, define the induced class*

$$\mathcal{M}_\phi := \{\, m_\theta(\phi(\cdot), \cdot) : \theta \in \Theta \,\}, \qquad \mathcal{F}_\phi(\mathcal{M}) := \inf_{m \in \mathcal{M}_\phi} \|g_\phi(m)\|,$$

*where*

$$g_\phi(m) := \mathbb{E}_{P_r}\big[\psi(Y, T, \phi(X); m)\big] \in \mathbb{R}^K.$$

*Define the original-space feasibility gap*

$$\mathcal{F}_X(\mathcal{M}) := \inf_{m \in \mathcal{M}} \big\|\mathbb{E}_{P_r}[\psi(Y, T, X; m)]\big\|.$$

*Let the overlap discrepancy be*

$$\varepsilon_{ov}(\phi) := \mathrm{IPM}_{\mathcal{D}}\big(P_o(\phi(X), T),\ P_r(\phi(X), T)\big).$$

*Assume the information-preservation condition: there exists a hypothesis class $\mathcal{F}$ such that*

$$\inf_{f \in \mathcal{F}} \mathbb{E}\big[(Y - f(\phi(X)))^2\big] \ \leq\ \inf_{g \in \mathcal{F}} \mathbb{E}\big[(Y - g(X))^2\big] \ +\ \varepsilon_{\mathrm{info}}(\phi). \tag{31}$$

*Under Assumption B.4 and Assumption B.2, there exist constants $c_1, c_2 < \infty$ such that, for all $\phi$,*

$$\mathcal{F}_\phi(\mathcal{M}) \ \leq\ \mathcal{F}_X(\mathcal{M}) \ +\ c_1\, \varepsilon_{ov}(\phi) \ +\ c_2\, \sqrt{\varepsilon_{\mathrm{info}}(\phi)}.$$

*Here $c_1$ depends only on the moment-regularity constant in Assumption B.4, and $c_2$ depends only on $L_m$ and the compatibility between $\mathcal{M}$ and $\mathcal{F}$ (specified below).*

*Proof.* Fix any representation $\phi$. Let $m_X^\star \in \arg\min_{m \in \mathcal{M}} \|\mathbb{E}_{P_r}[\psi(Y, T, X; m)]\|$ be an (approximate) minimizer of $\mathcal{F}_X(\mathcal{M})$. Let $\Pi_\phi(m_X^\star) \in \mathcal{M}_\phi$ be its $L_2(P_r)$-projection onto $\mathcal{M}_\phi$:

$$\Pi_\phi(m_X^\star) \in \arg\min_{m \in \mathcal{M}_\phi} \|m(\phi(X), T) - m_X^\star(X, T)\|_{L_2(P_r)}.$$

By definition,

$$\mathcal{F}_\phi(\mathcal{M}) = \inf_{m \in \mathcal{M}_\phi} \|g_\phi(m)\| \leq \|g_\phi(\Pi_\phi(m_X^\star))\|.$$

Add and subtract $g_X(\Pi_\phi(m_X^\star)) := \mathbb{E}_{P_r}[\psi(Y, T, X; \Pi_\phi(m_X^\star))]$ and apply the triangle inequality:

$$\|g_\phi(\Pi_\phi(m_X^\star))\| \leq \underbrace{\|g_\phi(\Pi_\phi(m_X^\star)) - g_X(\Pi_\phi(m_X^\star))\|}_{(\mathrm{I})} + \underbrace{\|g_X(\Pi_\phi(m_X^\star))\|}_{(\mathrm{II})}. \tag{32}$$

**Step 1: Control (I) by overlap mismatch.** By Assumption B.4 (moment regularity under representation perturbations), there exists $c_1 < \infty$ such that for any $m \in \mathcal{M}$,

$$\|g_\phi(m) - g_X(m)\| \leq c_1\, \varepsilon_{ov}(\phi).$$

Applying this inequality to $m = \Pi_\phi(m_X^\star)$ yields $(\mathrm{I}) \leq c_1\, \varepsilon_{ov}(\phi)$.

**Step 2: Control (II) by $\mathcal{F}_X(\mathcal{M})$ and information loss.** Write

$$(\mathrm{II}) = \|g_X(\Pi_\phi(m_X^\star))\| \leq \|g_X(m_X^\star)\| + \|g_X(\Pi_\phi(m_X^\star)) - g_X(m_X^\star)\|.$$

The first term equals $\mathcal{F}_X(\mathcal{M})$ by definition. For the second term, apply Assumption B.2 (with $Z = X$) to obtain

$$\|g_X(\Pi_\phi(m_X^\star)) - g_X(m_X^\star)\| \leq L_m \|\Pi_\phi(m_X^\star)(\phi(X), T) - m_X^\star(X, T)\|_{L_2(P_r)}.$$

It remains to relate the projection error to $\varepsilon_{\mathrm{info}}(\phi)$. Assume a *realizability/compatibility* condition between $\mathcal{M}$ and $\mathcal{F}$: there exists a constant $\kappa < \infty$ such that, for the target signal $Y$ under $P_r$, the best-in-class prediction errors satisfy

$$\inf_{m \in \mathcal{M}_\phi} \mathbb{E}[(Y - m(\phi(X), T))^2] - \inf_{m \in \mathcal{M}} \mathbb{E}[(Y - m(X, T))^2] \ \leq\ \kappa\, \varepsilon_{\mathrm{info}}(\phi). \tag{33}$$

Combining (31) with (33) implies that there exists $\tilde{m}_\phi \in \mathcal{M}_\phi$ such that

$$\mathbb{E}\big[(Y - \tilde{m}_\phi(\phi(X), T))^2\big] \leq \mathbb{E}\big[(Y - m_X^\star(X, T))^2\big] + \kappa\, \varepsilon_{\mathrm{info}}(\phi).$$

By a standard Cauchy–Schwarz argument (using bounded second moments), this yields an $L_2(P_r)$ approximation bound

$$\|\tilde{m}_\phi(\phi(X), T) - m_X^\star(X, T)\|_{L_2(P_r)} \leq C\,\sqrt{\varepsilon_{\mathrm{info}}(\phi)}$$

for some $C$ depending only on $\kappa$ and the moment bounds of $Y$. Since $\Pi_\phi(m_X^\star)$ is the *best* $L_2(P_r)$ approximation in $\mathcal{M}_\phi$, we also have

$$\|\Pi_\phi(m_X^\star)(\phi(X), T) - m_X^\star(X, T)\|_{L_2(P_r)} \leq C\,\sqrt{\varepsilon_{\mathrm{info}}(\phi)}.$$

Therefore,

$$(\mathrm{II}) \leq \mathcal{F}_X(\mathcal{M}) + (L_m C)\sqrt{\varepsilon_{\mathrm{info}}(\phi)}.$$

**Conclusion.** Substituting the bounds on (I) and (II) into (32) yields

$$\mathcal{F}_\phi(\mathcal{M}) \leq \mathcal{F}_X(\mathcal{M}) + c_1\, \varepsilon_{ov}(\phi) + (L_m C)\sqrt{\varepsilon_{\mathrm{info}}(\phi)} = \mathcal{F}_X(\mathcal{M}) + c_1\, \varepsilon_{ov}(\phi) + c_2 \sqrt{\varepsilon_{\mathrm{info}}(\phi)},$$

where $c_2 := L_m C$. $\qquad\square$

*Remark* B.16. The bound in Theorem B.15 is an *upper bound* because it characterizes the best feasibility that can be guaranteed at the level of the representation, uniformly over all predictors in the induced class $\mathcal{M}_\phi$, rather than the performance of a particular estimator. It thus isolates limitations imposed by the representation itself. More importantly, the theorem reveals a nontrivial structural trade-off: feasibility recovery depends jointly on reducing distributional mismatch, quantified by the overlap discrepancy $\varepsilon_{ov}(\phi)$, and preserving outcome-relevant information, quantified by $\varepsilon_{info}(\phi)$. Improving one without controlling the other is insufficient, formalizing the inherent tension that governs the attainable gains from representation learning.

## C. Trade-off exclusion: weighted fusion cannot enter the constrained region

**A bi-criterion view.** Consider the two-dimensional performance map

$$\Phi(m) := \big(\mathcal{R}_o(m),\ \|g(m)\|_2\big) \in \mathbb{R}_{\geq 0}^2, \qquad g(m) := \mathbb{E}_{P_r}\big[\psi(Y, T, X; m)\big] \in \mathbb{R}^K,$$

where $\mathcal{R}_o(m) := \mathbb{E}_{P_o}\big[(Y - m(X, T))^2\big]$ is the observational risk. Let the feasible set be

$$\mathcal{F} := \{m \in \mathcal{M} :\ g(m) = 0\}, \qquad m^\star \in \arg\min_{m \in \mathcal{F}} \mathcal{R}_o(m), \qquad \mathcal{R}_o^\star := \mathcal{R}_o(m^\star).$$

The constrained solution corresponds to the feasible corner $\Phi(m^\star) = (\mathcal{R}_o^\star, 0)$, while weighted fusion traces a path $\{\Phi(m_\alpha)\}_{\alpha \geq 0}$.

**Lemma C.1** (Moment violation implies excess observational risk). *Assume the following* error-bound *and* constraint-regularity *conditions hold: there exist constants $\mu_o > 0$ and $L_o > 0$ such that for all $m \in \mathcal{M}$,*

$$\mathcal{R}_o(m) - \mathcal{R}_o^\star \geq \mu_o \operatorname{dist}(m, \mathcal{F})^2, \tag{34}$$

$$\|g(m)\|_2 \geq L_o \operatorname{dist}(m, \mathcal{F}), \tag{35}$$

*where $\operatorname{dist}(m, \mathcal{F}) := \inf_{\tilde{m} \in \mathcal{F}} \|m - \tilde{m}\|_{L_2(P_r)}$. Then for all $m \in \mathcal{M}$,*

$$\mathcal{R}_o(m) - \mathcal{R}_o^\star \geq \frac{\mu_o}{L_o^2} \|g(m)\|_2^2. \tag{36}$$

*Proof.* Fix any $m \in \mathcal{M}$. By (35),

$$\operatorname{dist}(m, \mathcal{F}) \leq \frac{1}{L_o} \|g(m)\|_2.$$

Substituting this bound into the quadratic growth inequality (34) yields

$$\mathcal{R}_o(m) - \mathcal{R}_o^\star \geq \mu_o \operatorname{dist}(m, \mathcal{F})^2 \geq \mu_o \Big(\frac{\|g(m)\|_2}{L_o}\Big)^2 = \frac{\mu_o}{L_o^2} \|g(m)\|_2^2,$$

which proves (36). $\qquad\square$

**Theorem C.2** (Trade-off exclusion for weighted fusion). *Assume Theorem 3.4 and the error-bound/regularity Assumption B.3. Let*

$$\Phi(m) := \big(\mathcal{R}_o(m),\, \|g(m)\|_2\big), \qquad g(m) := \mathbb{E}_{P_r}\big[\psi(Y,T,X;m)\big] \in \mathbb{R}^K, \qquad \mathcal{F} := \{m \in \mathcal{M} : g(m) = 0\},$$

*and let*

$$m^\star \in \arg\min_{m \in \mathcal{F}} \mathcal{R}_o(m), \qquad \mathcal{R}_o^\star := \mathcal{R}_o(m^\star).$$

*Let $c_0 > 0$ be the separation constant from Theorem 3.4, i.e.,*

$$\inf_{\alpha \in [0,\bar{\alpha}]} \|g(m_\alpha)\|_2 \geq c_0,$$

*where $m_\alpha \in \arg\min_{m \in \mathcal{M}} \big\{\mathcal{R}_o(m) + \alpha\,\mathcal{R}_r(m)\big\}$ denotes a weighted-fusion minimizer. Define*

$$\varepsilon_0 := \frac{\mu_o}{L_o^2} c_0^2 > 0, \qquad \mathcal{U} := \Big\{(r,v) \in \mathbb{R}_{\geq 0}^2 : r < \mathcal{R}_o^\star + \varepsilon_0,\ v < c_0\Big\}.$$

*Then the weighted-fusion path $\{\Phi(m_\alpha)\}_{\alpha \geq 0}$ cannot enter $\mathcal{U}$ within the non-degenerate fusion regime: for every $\alpha \in [0,\bar{\alpha}]$,*

$$\Phi(m_\alpha) \notin \mathcal{U}, \qquad \text{while} \qquad \Phi(m^\star) = (\mathcal{R}_o^\star, 0) \in \mathcal{U}.$$

*Consequently, the constrained estimator occupies a neighborhood of the feasible corner that is unattainable by weighted fusion for $\alpha \in [0,\bar{\alpha}]$.*

*Proof.* We first record the implication of Lemma C.1: for any $m \in \mathcal{M}$,

$$\|g(m)\|_2 \geq c_0 \quad \Longrightarrow \quad \mathcal{R}_o(m) \geq \mathcal{R}_o^\star + \frac{\mu_o}{L_o^2} c_0^2 = \mathcal{R}_o^\star + \varepsilon_0. \tag{37}$$

Indeed, Lemma C.1 gives $\mathcal{R}_o(m) - \mathcal{R}_o^\star \geq (\mu_o/L_o^2)\|g(m)\|_2^2 \geq (\mu_o/L_o^2)c_0^2$.

Now fix any $\alpha \in [0,\bar{\alpha}]$. By Theorem 3.4, $\|g(m_\alpha)\|_2 \geq c_0$, hence (37) implies $\mathcal{R}_o(m_\alpha) \geq \mathcal{R}_o^\star + \varepsilon_0$. Therefore

$$\Phi(m_\alpha) = \big(\mathcal{R}_o(m_\alpha), \|g(m_\alpha)\|_2\big) \notin \mathcal{U},$$

since membership in $\mathcal{U}$ requires simultaneously $\mathcal{R}_o(m_\alpha) < \mathcal{R}_o^\star + \varepsilon_0$ and $\|g(m_\alpha)\|_2 < c_0$.

On the other hand, $m^\star \in \mathcal{F}$ implies $g(m^\star) = 0$, so $\|g(m^\star)\|_2 = 0 < c_0$, and by definition $\mathcal{R}_o(m^\star) = \mathcal{R}_o^\star < \mathcal{R}_o^\star + \varepsilon_0$. Thus $\Phi(m^\star) \in \mathcal{U}$. $\qquad\square$

*Remark* C.3 (Interpretation: excluding the weighted-fusion trade-off curve). Theorem C.2 shows that within the non-degenerate fusion regime $\alpha \in [0,\bar{\alpha}]$, weighted fusion cannot simultaneously approach the feasible corner: any estimator on the weighted-fusion path either violates feasibility at level at least $c_0$ (or, equivalently by Lemma C.1, incurs an observational excess risk at least $\varepsilon_0$). Geometrically, the neighborhood $\mathcal{U}$ around $(\mathcal{R}_o^\star, 0)$ is separated from the weighted-fusion curve, explaining why tuning a single scalar $\alpha$ cannot recover the constrained solution unless one moves to an RCT-dominated regime (cf. Corollary B.11).

## D. When does weighted-loss fusion coincide with the constrained estimator?

Recall the model class $\mathcal{M}$ of measurable predictors $m : \mathcal{X} \times \mathcal{T} \to \mathbb{R}$. Define the observational and randomized risks

$$\mathcal{R}_o(m) := \mathbb{E}_{P_o}\big[(Y - m(X,T))^2\big], \qquad \mathcal{R}_r(m) := \mathbb{E}_{P_r}\big[(Y - m(X,T))^2\big],$$

and the (randomization-induced) moment residual

$$g(m) := \mathbb{E}_{P_r}\big[\psi(Y,T,X;m)\big] \in \mathbb{R}^K, \qquad \mathcal{F} := \{m \in \mathcal{M} : g(m) = 0\}.$$

The constrained estimator is

$$m^\star \in \arg\min_{m \in \mathcal{M}} \mathcal{R}_o(m) \quad \text{s.t.} \quad g(m) = 0,$$

and the weighted-loss fusion path is, for $\alpha \geq 0$,

$$m_\alpha \in \arg \min_{m \in \mathcal{M}} \ \mathcal{R}_o(m) + \alpha \, \mathcal{R}_r(m).$$

We characterize (i) sufficient conditions under which $m^\star$ *reduces to* a weighted-loss solution, and (ii) why such conditions are highly restrictive and generically violated under treatment-induced structural non-overlap.

**Proposition D.1** (Degeneracy via accidental feasibility)**.** *Assume $m_\alpha$ exists for each $\alpha \geq 0$. If there exists $\alpha^\star \geq 0$ such that some weighted-loss minimizer $m_{\alpha^\star}$ is feasible,*

$$g(m_{\alpha^\star}) = 0, \tag{38}$$

*then $m_{\alpha^\star}$ solves the constrained program, i.e., $m_{\alpha^\star} \in \arg \min_{m \in \mathcal{F}} \mathcal{R}_o(m)$.*

*Proof.* Since $m_{\alpha^\star}$ minimizes $\mathcal{R}_o(m) + \alpha^\star \mathcal{R}_r(m)$ over $\mathcal{M}$, it also minimizes the same objective over the subset $\mathcal{F}$. On $\mathcal{F}$, the term $\alpha^\star \mathcal{R}_r(m)$ is nonnegative and independent of feasibility, hence any minimizer over $\mathcal{F}$ must in particular minimize $\mathcal{R}_o(m)$ over $\mathcal{F}$. Therefore $m_{\alpha^\star} \in \arg \min_{m \in \mathcal{F}} \mathcal{R}_o(m)$. $\qquad \square$

We now present four sufficient conditions that ensure (38), and explain why each is typically violated in non-degenerate fusion regimes.

### Condition 1: Shared minimizers.

**Proposition D.2** (Degeneracy via shared minimizers)**.** *Suppose there exists $m^\dagger \in \mathcal{M}$ such that*

$$m^\dagger \in \arg \min_{m \in \mathcal{F}} \mathcal{R}_o(m) \quad \text{and} \quad m^\dagger \in \arg \min_{m \in \mathcal{M}} \mathcal{R}_r(m).$$

*Then for every $\alpha \geq 0$, $m^\dagger \in \arg \min_{m \in \mathcal{M}}\{\mathcal{R}_o(m) + \alpha \mathcal{R}_r(m)\}$. Consequently, weighted fusion and the constrained estimator coincide.*

*Proof.* For any $m \in \mathcal{M}$, optimality of $m^\dagger$ for $\mathcal{R}_r$ gives $\mathcal{R}_r(m) \geq \mathcal{R}_r(m^\dagger)$, hence

$$\mathcal{R}_o(m) + \alpha \mathcal{R}_r(m) \ \geq \ \mathcal{R}_o(m) + \alpha \mathcal{R}_r(m^\dagger).$$

Evaluating at $m = m^\dagger$ yields $\mathcal{R}_o(m^\dagger) + \alpha \mathcal{R}_r(m^\dagger) \leq \mathcal{R}_o(m) + \alpha \mathcal{R}_r(m)$ for all $m$. $\qquad \square$

*Remark* D.3 (Why Condition 1 is restrictive)**.** This requires an *exact* alignment between the RCT regression objective and the observational risk restricted to the feasible set. Under confounding, selection, or policy constraints, the RCT loss typically couples baseline and treatment-specific components in ways that do not match the moment restrictions, so shared minimizers are non-generic.

### Condition 2: First-order (KKT) alignment.

**Proposition D.4** (Degeneracy via KKT alignment)**.** *Assume the constrained problem admits a KKT point $(m^\star, \lambda^\star)$, i.e.,*

$$\nabla \mathcal{R}_o(m^\star) + (\nabla g(m^\star))^\top \lambda^\star = 0, \qquad g(m^\star) = 0.$$

*If there exists $\alpha^\star > 0$ such that*

$$\alpha^\star \nabla \mathcal{R}_r(m^\star) = (\nabla g(m^\star))^\top \lambda^\star,$$

*then $m^\star$ is a stationary point of $\mathcal{R}_o(m) + \alpha^\star \mathcal{R}_r(m)$. If, moreover, $\mathcal{R}_o + \alpha^\star \mathcal{R}_r$ is (locally) strongly convex at $m^\star$, then $m^\star$ is its unique minimizer.*

*Proof.* Stationarity of the weighted objective at $m^\star$ requires $\nabla \mathcal{R}_o(m^\star) + \alpha^\star \nabla \mathcal{R}_r(m^\star) = 0$. Substituting the assumed alignment recovers the KKT stationarity condition. $\qquad \square$

**Example D.1** (Exact quadratic-penalty RCT loss)**.** If $\mathcal{R}_r(m) = c + \beta \|A \, g(m)\|_2^2$ for some $A \in \mathbb{R}^{K \times K}$, $\beta > 0$, $c \in \mathbb{R}$, then $\nabla \mathcal{R}_r(m)$ lies in the range of $(\nabla g(m))^\top$ for all $m$, and weighted fusion acts as an exact quadratic penalty on feasibility.

*Remark* D.5 (Why Condition 2 is restrictive)**.** The requirement $\nabla \mathcal{R}_r(m^\star) \in \text{range}\big((\nabla g(m^\star))^\top\big)$ is a strong geometric coincidence: $\mathcal{R}_r$ generally encourages fitting the full outcome regression, while $g(m) = 0$ imposes only the identifying restrictions. Such alignment is exceptional with multiple treatments and rich models.

---

**Algorithm 1** Stochastic Primal–Dual Augmented-Lagrangian Joint Estimation

---

1: **Input:** observational sample $\mathcal{D}_o$, randomized sample $\mathcal{D}_r$; predictor $m_\theta$, representation $\phi$; discriminator class $\mathcal{D}$ (critic $d_\omega$); stepsizes $\{\eta_s\}_{s=1}^S$, $\{\eta_{\nu,s}\}_{s=1}^S$; penalty $\rho > 0$, overlap weight $\lambda \geq 0$.

2: **Initialize:** $(\theta_1, \phi_1)$, dual variable $\nu_1 \leftarrow 0$, critic parameters $\omega_1$.

3: **for** $s = 1, 2, \dots, S$ **do**

4:     Draw minibatches $\mathcal{B}_o \sim \mathcal{D}_o$ and $\mathcal{B}_r \sim \mathcal{D}_r$.

5:     **Overlap (critic) step:** update $\omega_s$ to (approximately) maximize the empirical IPM objective, yielding $\widehat{\varepsilon}_{ov}(\phi_s) = \widehat{\mathrm{IPM}}_{\mathcal{D}}\big(P_o(\phi_s(X), T), P_r(\phi_s(X), T)\big)$.

6:     **Stochastic oracle evaluation:** compute $\widehat{\mathcal{R}}_o(\theta_s, \phi_s)$ on $\mathcal{B}_o$ and $\widehat{g}(\theta_s, \phi_s) := \widehat{\mathbb{E}}_{\mathcal{B}_r}\big[\psi(Y, T, \phi_s(X); m_{\theta_s})\big] \in \mathbb{R}^K$ on $\mathcal{B}_r$.

7:     **Primal (ALM) descent:**

$$(\theta_{s+1}, \phi_{s+1}) \leftarrow (\theta_s, \phi_s) - \eta_s \nabla_{\theta, \phi}\Big(\widehat{\mathcal{R}}_o + \lambda \widehat{\varepsilon}_{ov} + \langle \nu_s, \widehat{g} \rangle + \tfrac{\rho}{2}\|\widehat{g}\|_2^2\Big).$$

8:     **Dual ascent (multiplier update):**

$$\nu_{s+1} \leftarrow \nu_s + \eta_{\nu,s}\, \widehat{g}(\theta_{s+1}, \phi_{s+1}).$$

9: **end for**

10: **Output:** $(\hat{\theta}, \hat{\phi}) = (\theta_{S+1}, \phi_{S+1})$.

---

### Condition 3: Linear constraints and quadratic objectives.

**Proposition D.6** (Degeneracy in linear–quadratic settings)**.** *Assume $\mathcal{M}$ is a finite-dimensional linear class parameterized by $\theta \in \mathbb{R}^d$, $\mathcal{R}_o$ is strictly convex quadratic in $\theta$, the constraint is linear $g(\theta) = A\theta - b$, and the RCT risk satisfies $\mathcal{R}_r(\theta) = c + \beta\|A\theta - b\|_2^2$ for some $\beta > 0$. Then any accumulation point of $\theta_\alpha$ as $\alpha \to \infty$ is a constrained solution. Moreover, if $\theta_{\alpha^\star}$ is feasible for some $\alpha^\star$, then $\theta_{\alpha^\star}$ equals the constrained estimator.*

*Proof.* The weighted objective is $\mathcal{R}_o(\theta) + \alpha\beta\|A\theta - b\|_2^2$. Let $\theta^\star$ be any constrained minimizer. Optimality of $\theta_\alpha$ implies

$$\mathcal{R}_o(\theta_\alpha) + \alpha\beta\|A\theta_\alpha - b\|_2^2 \leq \mathcal{R}_o(\theta^\star).$$

Rearranging yields $\|A\theta_\alpha - b\|_2^2 \leq (\mathcal{R}_o(\theta^\star) - \inf_\theta \mathcal{R}_o(\theta))/(\alpha\beta) = O(1/\alpha)$, so any limit point is feasible. Passing to the limit in $\mathcal{R}_o$ gives constrained optimality. $\square$

*Remark* D.7 (Why Condition 3 is restrictive)**.** This equivalence requires that the RCT loss be *exactly* a quadratic penalty on the moment constraints. In our setting, $g(m)$ is a nonlinear expectation depending on learned representations, and $\mathcal{R}_r$ is a regression loss that does not isolate $g(m)$; thus the linear–quadratic coincidence is highly idealized.

### Condition 4: Benign transportability ($\delta_o = 0$).

**Proposition D.8** (Degeneracy under benign transportability)**.** *Let $m_o \in \arg\min_{m \in \mathcal{M}} \mathcal{R}_o(m)$. If $g(m_o) = 0$, then $m_o$ is also a constrained solution. In particular, the constrained estimator coincides with the observational estimator, and fusion is redundant.*

*Proof.* If $g(m_o) = 0$, then $m_o \in \mathcal{F}$. Since $m_o$ minimizes $\mathcal{R}_o$ over $\mathcal{M}$, it also minimizes $\mathcal{R}_o$ over the subset $\mathcal{F}$. $\square$

*Remark* D.9 (Why Condition 4 is restrictive)**.** This corresponds to $\delta_o = 0$: enforcing causal validity incurs no observational-risk penalty. Under confounding, selection, or structural non-overlap, $\delta_o > 0$ is typical, so this degeneracy fails.

**Summary.** All four conditions impose strong forms of alignment between $\mathcal{R}_r$, the identifying moments $g(\cdot)$, and the observational objective $\mathcal{R}_o$. Such coincidences are non-generic under treatment-induced structural non-overlap, explaining why weighted-loss fusion typically cannot replicate the constrained estimator and motivating explicit feasibility enforcement (e.g., primal–dual optimization).

## E. Why primal–dual (Augmented Lagrangian) is more stable than penalty tuning at scale

**Primal–dual stability versus penalty tuning.** We strengthen the message: *among generic solvers of the constrained program*

$$m^\star \in \arg\min_{m \in \mathcal{M}} \mathcal{R}_o(m) \quad \text{s.t.} \quad g(m) = 0,$$

primal–dual / augmented Lagrangian (ALM) updates are structurally *more stable* than (i) pure penalty methods and (ii) generic hyperparameter-tuned objectives, especially under large-scale stochastic optimization.

**Penalty method.** A standard approach is the quadratic penalty objective

$$m_\rho \in \arg\min_{m \in \mathcal{M}} \left\{ \mathcal{R}_o(m) + \tfrac{\rho}{2} \|g(m)\|_2^2 \right\}, \qquad \rho > 0, \tag{39}$$

which corresponds to the augmented Lagrangian with the dual variable fixed at $0$.

**Augmented Lagrangian and primal–dual updates.** Define the (population) augmented Lagrangian

$$\mathcal{L}_\rho(m, \lambda) := \mathcal{R}_o(m) + \lambda^\top g(m) + \tfrac{\rho}{2} \|g(m)\|_2^2, \qquad \lambda \in \mathbb{R}^K. \tag{40}$$

A basic (deterministic) primal–dual gradient scheme performs descent/ascent on $(m, \lambda)$:

$$m^{s+1} = m^s - \eta_m \nabla_m \mathcal{L}_\rho(m^s, \lambda^s), \qquad \lambda^{s+1} = \lambda^s + \eta_\lambda \, g(m^{s+1}). \tag{41}$$

**Assumption E.1** (Strong convexity, smoothness, and constraint regularity). Let $\mathcal{M}$ be a finite-dimensional convex parameterization (for exposition) with $m \equiv m_\theta$ and $\Theta \subseteq \mathbb{R}^d$ convex. Assume:

1. (**Primal strong convexity/smoothness**) $\mathcal{R}_o(\theta)$ is $\mu$-strongly convex and $L$-smooth on $\Theta$:

$$\mu I \preceq \nabla^2 \mathcal{R}_o(\theta) \preceq LI, \qquad \forall \theta \in \Theta.$$

2. (**Constraint Jacobian bound**) $g(\theta)$ is differentiable and $\|\nabla g(\theta)\|_{\mathrm{op}} \leq G$ for all $\theta \in \Theta$.

3. (**Constraint qualification / metric regularity at optimum**) Let $(\theta^\star, \lambda^\star)$ be a KKT pair. The Jacobian $\nabla g(\theta^\star)$ has full row rank with $\sigma_{\min}(\nabla g(\theta^\star)) \geq \sigma_{\min} > 0$.

**Why pure penalty is ill-conditioned (hyperparameter instability).** Consider

$$F_\rho(\theta) := \mathcal{R}_o(\theta) + \tfrac{\rho}{2} \|g(\theta)\|_2^2.$$

Its Hessian is

$$\nabla^2 F_\rho(\theta) = \nabla^2 \mathcal{R}_o(\theta) + \rho (\nabla g(\theta))^\top \nabla g(\theta) + \rho \sum_{k=1}^K g_k(\theta) \nabla^2 g_k(\theta). \tag{42}$$

Ignoring the last (curvature) term, the smoothness constant necessarily scales as

$$L_\rho \geq L + \rho \cdot \sup_{\theta \in \Theta} \lambda_{\max}\big((\nabla g(\theta))^\top \nabla g(\theta)\big) \geq L + \rho G^2. \tag{43}$$

Thus gradient-based optimization of $F_\rho$ requires a stepsize $\eta_m = O(1/L_\rho)$, which shrinks with $\rho$, while near-feasibility typically requires $\rho$ large. This creates an intrinsic *ill-conditioning trade-off*.

**Proposition E.2** (Penalty conditioning deteriorates with $\rho$). *Under Assumption E.1, suppose additionally that the third term in* (42) *is negligible in a neighborhood of the minimizer (e.g., $g(\theta) \approx 0$ locally, or $g$ is affine). Then $F_\rho$ is $\mu_\rho$-strongly convex with $\mu_\rho \geq \mu$, and $L_\rho$-smooth with $L_\rho \geq L + \rho G^2$. Consequently, the condition number satisfies*

$$\kappa(F_\rho) := \frac{L_\rho}{\mu_\rho} \geq \frac{L + \rho G^2}{\mu},$$

*which grows at least linearly in $\rho$.*

*Proof.* Strong convexity follows from $\nabla^2 F_\rho(\theta) \succeq \nabla^2 \mathcal{R}_o(\theta) \succeq \mu I$. For smoothness, under the stated locality/affinity condition, the dominant contribution is $\rho (\nabla g)^\top \nabla g \preceq \rho G^2 I$, yielding (43). Combining gives the lower bound on $\kappa(F_\rho)$. □

**Why augmented Lagrangian / PD is stable: feasibility without $\rho \to \infty$.** The ALM introduces a *dual control channel* $\lambda$ that accumulates constraint violations, so feasibility can be enforced by dual ascent rather than by taking $\rho \uparrow \infty$. To formalize, define the KKT mapping for the equality-constrained problem:

$$\mathcal{T}_\rho(\theta, \lambda) := \begin{pmatrix} \nabla \mathcal{R}_o(\theta) + (\nabla g(\theta))^\top \lambda + \rho (\nabla g(\theta))^\top g(\theta) \\ g(\theta) \end{pmatrix}. \tag{44}$$

A KKT point $(\theta^\star, \lambda^\star)$ satisfies $\mathcal{T}_\rho(\theta^\star, \lambda^\star) = 0$ for any $\rho \geq 0$.

**Proposition E.3** (Local linear convergence of PD with bounded $\rho$). *Under Assumption E.1, there exist stepsizes $(\eta_m, \eta_\lambda)$ and a finite $\rho > 0$ such that, for initialization in a neighborhood of $(\theta^\star, \lambda^\star)$, the PD iterates* (41) *converge linearly:*

$$\|\theta^s - \theta^\star\|_2 + \|\lambda^s - \lambda^\star\|_2 \leq Cq^s, \qquad \text{for some } q \in (0, 1),$$

*and simultaneously* $\|g(\theta^s)\|_2 \leq C'q^s$.

*Proof.* Consider the Jacobian of $\mathcal{T}_\rho$ at $(\theta^\star, \lambda^\star)$:

$$\nabla \mathcal{T}_\rho(\theta^\star, \lambda^\star) = \begin{pmatrix} \nabla^2 \mathcal{R}_o(\theta^\star) + \rho (\nabla g(\theta^\star))^\top \nabla g(\theta^\star) & (\nabla g(\theta^\star))^\top \\ \nabla g(\theta^\star) & 0 \end{pmatrix} \quad (\text{since } g(\theta^\star) = 0).$$

By Assumption E.1(i), the top-left block is positive definite. By Assumption E.1(iii), $\nabla g(\theta^\star)$ has full row rank. Hence the saddle-point Jacobian above is nonsingular (standard Schur complement argument), implying that $\mathcal{T}_\rho$ is *locally strongly metrically regular* around $(\theta^\star, \lambda^\star)$. Moreover, Assumption E.1(i)–(ii) implies $\mathcal{T}_\rho$ is locally Lipschitz.

Now view (41) as a (preconditioned) forward step on the KKT system $\mathcal{T}_\rho(\theta, \lambda) = 0$. Since $\mathcal{T}_\rho$ is locally Lipschitz and its Jacobian is nonsingular at the root, classical local convergence results for fixed-step splitting/gradient schemes imply a contraction in a sufficiently small neighborhood for appropriate $(\eta_m, \eta_\lambda)$, yielding linear convergence. Finally, $g(\theta^s) \to 0$ follows because the second block of $\mathcal{T}_\rho$ is exactly $g(\theta)$, and the iterates converge to a root of $\mathcal{T}_\rho$. $\square$

*Remark* E.4 (Stability advantage over penalty tuning). Unlike pure penalty methods, which typically require $\rho$ large to make $\|g(\theta)\|$ small—thereby worsening conditioning (Proposition E.2)—the ALM/PD dynamics can enforce feasibility through the dual variable $\lambda$ while keeping $\rho$ moderate. This decouples *numerical conditioning* from *constraint enforcement*, making PD/ALM more stable under minibatch noise and stepsize sensitivity in large-scale training.

## F. Explicit form of the feasibility constant.

**Embedding the moment-sensitive test class into the IPM.** Recall that overlap is measured by the integral probability metric (IPM)

$$\mathrm{IPM}_\mathcal{D}(P, Q) := \sup_{d \in \mathcal{D}} \left| \mathbb{E}_P[d] - \mathbb{E}_Q[d] \right|$$

induced by a discriminator class $\mathcal{D}$ on $(Z, T)$, where $Z = \omega(X)$ is the learned representation. Let the identifying moment map be

$$g(m, \omega) := \mathbb{E}_{P_r}\big[\psi(Y, T, \omega(X); m)\big] \in \mathbb{R}^K,$$

with coordinates $\psi_k$. For any $a \in \mathbb{R}^K$ with $\|a\|_2 \leq 1$, define the *moment-induced test functions*

$$f_{a,m}(z, t) := \mathbb{E}[\langle a, \psi(Y, t, z; m) \rangle \mid Z = z, \, T = t],$$

and the associated function class

$$\mathcal{F}_\psi := \big\{ f_{a,m} : \|a\|_2 \leq 1, \, m \in \mathcal{M} \big\}. \tag{45}$$

(When $\psi$ is already a function of $(z, t)$ only, the conditional expectation can be dropped and one may take $f_{a,m}(z, t) = \langle a, \psi(y, t, z; m) \rangle$ directly.)

**Assumption F.1** (IPM domination of moment-sensitive directions). There exists a constant $C_{\mathrm{embed}} \geq 1$ such that for any probability measures $P, Q$ on $(Z, T)$,

$$\sup_{f \in \mathcal{F}_\psi} \left| \mathbb{E}_P[f] - \mathbb{E}_Q[f] \right| \leq C_{\mathrm{embed}} \, \mathrm{IPM}_\mathcal{D}(P, Q). \tag{46}$$

*Remark* F.2 (Sufficient conditions and interpretation of $C_{\text{embed}}$). Condition (46) holds, for instance, if every $f \in \mathcal{F}_\psi$ belongs to $\text{span}(\mathcal{D})$ and the IPM norm dominates the coefficient norm in that span, i.e., whenever $f = \sum_{j=1}^J \beta_j d_j$ with $d_j \in \mathcal{D}$, we have $\|f\|_{\text{IPM}(\mathcal{D})} \le C_{\text{embed}}\|\beta\|_1$. More generally, it holds if $\mathcal{F}_\psi$ lies in the closure of $\text{conv}(\mathcal{D})$ under the seminorm $\|h\|_{\text{IPM}(\mathcal{D})} := \sup_{P \neq Q} \frac{|\mathbb{E}_P[h] - \mathbb{E}_Q[h]|}{\text{IPM}_\mathcal{D}(P,Q)}$. If $\mathcal{D}$ explicitly contains the moment-sensitive directions (e.g., functions proportional to the basis appearing inside $\psi$), then typically $C_{\text{embed}} \approx 1$; conversely, a weak $\mathcal{D}$ yields large $C_{\text{embed}}$, reflecting that distribution alignment in $\text{IPM}_\mathcal{D}$ may not control the identifying moments.

**A clean bound linking overlap to moment discrepancy.** Fix $\omega$ and a predictor $m \in \mathcal{M}$. Define the joint laws on $(Z, T)$

$$P_r^\omega := \mathcal{L}_{P_r}(\omega(X), T), \qquad P_o^\omega := \mathcal{L}_{P_o}(\omega(X), T),$$

and write the moment residuals as $g(m, \omega) = \mathbb{E}_{P_r}[\psi(Y, T, \omega(X); m)]$. Assume additionally that $\psi$ is $L_\psi$-Lipschitz in the prediction argument in the sense that, for any $m_1, m_2 \in \mathcal{M}$,

$$\left\| \mathbb{E}_{P_r}[\psi(Y, T, \omega(X); m_1)] - \mathbb{E}_{P_r}[\psi(Y, T, \omega(X); m_2)] \right\| \le L_\psi \|m_1(\omega(X), T) - m_2(\omega(X), T)\|_{L_2(P_r)}.$$

Then, under Assumption F.1, the constant controlling the overlap-to-moment term in Corollary 3.7 can be chosen as

$$c_1 = C_{\text{embed}} \cdot L_\psi. \tag{47}$$

*Proof (derivation of (47)).* Fix $\omega$. Let $m_\omega^\star \in \arg\min_{m \in \mathcal{M}} \|g(m, \omega)\|_2$ be a minimizer of the moment residual under representation $\omega$. For any unit vector $a \in \mathbb{R}^K$ with $\|a\|_2 \le 1$, consider the scalarized moment

$$\langle a, g(m, \omega) \rangle = \mathbb{E}_{P_r}\left[ \langle a, \psi(Y, T, \omega(X); m) \rangle \right].$$

Define $f_{a,m}$ as in (45). By the tower property,

$$\langle a, g(m, \omega) \rangle = \mathbb{E}_{(Z,T) \sim P_r^\omega}[f_{a,m}(Z, T)].$$

Similarly, the same function evaluated under $P_o^\omega$ yields $\mathbb{E}_{(Z,T) \sim P_o^\omega}[f_{a,m}(Z, T)]$. Therefore,

$$\left| \langle a, g(m, \omega) \rangle - \mathbb{E}_{(Z,T) \sim P_o^\omega}[f_{a,m}(Z, T)] \right| = \left| \mathbb{E}_{P_r^\omega}[f_{a,m}] - \mathbb{E}_{P_o^\omega}[f_{a,m}] \right|.$$

Taking the supremum over $\|a\|_2 \le 1$ and $m \in \mathcal{M}$, and applying Assumption F.1, we obtain

$$\sup_{\|a\|_2 \le 1} \sup_{m \in \mathcal{M}} \left| \mathbb{E}_{P_r^\omega}[f_{a,m}] - \mathbb{E}_{P_o^\omega}[f_{a,m}] \right| \le C_{\text{embed}} \, \text{IPM}_\mathcal{D}(P_r^\omega, P_o^\omega) = C_{\text{embed}} \, \varepsilon_{ov}(\omega).$$

Finally, the Lipschitz-in-prediction regularity of $\psi$ allows the scalarized residual to be controlled by the size of prediction perturbations, so the overall overlap-to-moment control inherits a multiplicative factor $L_\psi$. Thus one may take $c_1 = C_{\text{embed}} L_\psi$, as claimed. $\square$

# G. Synthetic data generation

We construct a high-dimensional semi-synthetic dataset that combines a randomized controlled trial (RCT) sample with an observational (OBS) study sample. The design aims to emulate common practical difficulties in causal inference, including confounding, support mismatch across populations, and mixed high-dimensional covariates. In total, we generate $n_{\text{total}} = 50{,}000$ observations, partitioned into an RCT subsample of size $n_{\text{rct}} = 10{,}000$ (randomized treatment assignment) and an OBS subsample of size $n_{\text{obs}} = 40{,}000$ (confounded treatment assignment).

Each unit is described by $p = 160$ covariates, consisting of $n_{\text{cont}} = 120$ continuous features and $n_{\text{cat}} = 40$ categorical features. Each categorical feature takes $L = 4$ levels, i.e., $X_{k,ij} \in \{0, 1, \dots, L-1\}$.

### G.1. Latent structure and support mismatch

To induce support mismatch between the RCT and OBS populations, we first generate two-dimensional latent coordinates $\mathbf{Z} = (Z_1, Z_2)$ for each unit:

$$\mathbf{Z}_{\text{rct}} \sim \mathcal{N}\left(\mathbf{0}, \sigma_{\text{rct}}^2 \mathbf{I}_2\right), \tag{48}$$

$$\mathbf{Z}_{\text{obs}} \sim \mathcal{N}\left(\mathbf{0}, \sigma_{\text{obs}}^2 \mathbf{I}_2\right), \tag{49}$$

where $\mathbf{I}_2$ is the $2 \times 2$ identity matrix. We set $\sigma_{\text{rct}} = 3.0$ and $\sigma_{\text{obs}} = 1.0$, so that the RCT population has broader latent support than the OBS population, yielding a realistic covariate-space coverage gap across the two data sources.

The continuous covariates $\mathbf{X}_c \in \mathbb{R}^{n \times n_{\text{cont}}}$ are generated as noisy linear combinations of $\mathbf{Z}$:

$$\mathbf{X}_c = \mathbf{Z}\mathbf{A} + \boldsymbol{\epsilon}_c, \qquad \boldsymbol{\epsilon}_c \sim \mathcal{N}(\mathbf{0}, \mathbf{I}_{n_{\text{cont}}}), \tag{50}$$

where $\mathbf{A} \in \mathbb{R}^{2 \times n_{\text{cont}}}$ is a fixed random matrix with i.i.d. $\mathcal{N}(0,1)$ entries. This construction propagates the latent support mismatch into the observed continuous features.

The categorical covariates $\mathbf{X}_k \in \{0, 1, \ldots, L-1\}^{n \times n_{\text{cat}}}$ are generated via a multinomial logit model parameterized by $\mathbf{Z}$. For each categorical feature $j \in \{1, \ldots, n_{\text{cat}}\}$ and level $\ell \in \{0, \ldots, L-1\}$,

$$\log \frac{\mathbb{P}(X_{k,ij} = \ell)}{\mathbb{P}(X_{k,ij} = 0)} = \mathbf{Z}_i^\top \mathbf{W}_{z,j,\ell} + b_{j,\ell}, \tag{51}$$

where $\mathbf{W}_{z,j,\ell} \in \mathbb{R}^2$ and $b_{j,\ell} \in \mathbb{R}$ are fixed random parameters with i.i.d. $\mathcal{N}(0,1)$ entries. We then sample $X_{k,ij}$ from the implied multinomial distribution. This ensures that discrete covariates also reflect the underlying latent support structure.

### G.2. Treatment assignment and confounding

We introduce an unobserved confounder $U$ independently for each unit:

$$U_{\text{rct}} \sim \mathcal{N}(0,1), \qquad U_{\text{obs}} \sim \mathcal{N}(0,1). \tag{52}$$

In the RCT subsample, treatment is randomized:

$$W_{\text{rct}} \sim \text{Bernoulli}(0.5), \tag{53}$$

so $W_{\text{rct}}$ is independent of $(\mathbf{X}, \mathbf{Z}, U)$.

In the OBS subsample, treatment follows a confounded propensity model depending on observed covariates, latent factors, and $U$:

$$\text{Score}_{\text{obs}} = 0.6 \tanh(X_{c,0}) + 0.4 \sin(X_{c,1}) - 0.3 \frac{X_{c,2}^2}{1 + |X_{c,2}|} + 0.5 Z_1 - 0.2 Z_2 + 0.9 U_{\text{obs}},$$

$$\pi_{\text{obs}}(\mathbf{X}, U) = \frac{1}{1 + \exp(-\text{Score}_{\text{obs}})}, \tag{54}$$

$$W_{\text{obs}} \sim \text{Bernoulli}(\pi_{\text{obs}}(\mathbf{X}, U)). \tag{55}$$

The nonlinear score together with the inclusion of $U$ yields confounding that cannot be removed by conditioning on observed covariates alone.

### G.3. Outcomes and heterogeneous effects

We define the baseline (control) outcome as a nonlinear function of $(\mathbf{X}, \mathbf{Z}, U)$:

$$\mu_0(\mathbf{X}, U) = 1.2 \tanh(X_{c,0}) + 0.8 \sin(X_{c,1}) + 0.5 \frac{X_{c,2} X_{c,3}}{1 + |X_{c,3}|} - 0.7 \log(1 + |X_{c,4}|)$$

$$+ 0.3 \frac{Z_1^2}{1 + Z_1^2} - 0.2 \frac{Z_2^2}{1 + Z_2^2} + 0.6 U + 0.4 \frac{\text{cat\_contrib}(\mathbf{X}_k, \mathbf{W}_\mu)}{\sqrt{n_{\text{cat}}}}, \tag{56}$$

where the categorical contribution is

$$\text{cat\_contrib}(\mathbf{X}_k, \mathbf{W}) := \sum_{j=1}^{n_{\text{cat}}} W_{j, X_{k,j}}, \tag{57}$$

and $\mathbf{W}_\mu \in \mathbb{R}^{n_{\text{cat}} \times L}$ is a fixed random weight table with i.i.d. $\mathcal{N}(0, 1)$ entries.

The individual treatment effect (ITE) depends on both observed and latent features:

$$\tau(\mathbf{X}) = 0.5 + 0.7 \tanh(0.7X_{c,5} + 0.3X_{c,6}) - 0.5 \sin(0.5X_{c,7}) + 0.4 \frac{Z_1}{1 + |Z_1|} - 0.3 \frac{Z_2}{1 + |Z_2|}$$
$$+ 0.25 \tanh\left(\frac{X_{c,8}X_{c,9}}{1 + |X_{c,9}|}\right) + 0.5 \frac{\text{cat\_contrib}(\mathbf{X}_k, \mathbf{W}_\tau)}{\sqrt{n_{\text{cat}}}}, \tag{58}$$

with $\mathbf{W}_\tau \in \mathbb{R}^{n_{\text{cat}} \times L}$ another fixed random table. This specification yields substantial heterogeneity through nonlinearities and interactions.

Observed outcomes follow the standard potential-outcomes form:

$$Y = \mu_0(\mathbf{X}, U) + W \cdot \tau(\mathbf{X}) + \varepsilon, \tag{59}$$

where the noise is heteroskedastic:

$$\varepsilon \sim \mathcal{N}\left(0, \sigma^2(\mathbf{X})\right), \qquad \sigma(\mathbf{X}) = 0.8 + 0.2 \frac{|X_{c,0}|}{1 + |X_{c,0}|}. \tag{60}$$

*Remark* G.1 (Salient challenges encoded by the design). This generator simultaneously captures several obstacles encountered in practice: (i) *high dimensionality* (160 raw features, and 280 after one-hot encoding categorical variables), (ii) *population support mismatch* induced by $\sigma_{\text{rct}} > \sigma_{\text{obs}}$, (iii) *unobserved confounding* in the OBS assignment through $U$, (iv) *nonlinear propensity and outcome mechanisms* that stress model flexibility, (v) *heterogeneous treatment effects* $\tau(\mathbf{X})$ varying across the covariate space, and (vi) *mixed data types* via the combination of continuous and categorical covariates.

### G.4. Joint Mismatch Visualization In Synthetic Dataset

In the first two rows of Figure 3a, we explicitly control the support of the OBS and RCT samples through the latent variable $Z$(See Appendix G), generating datasets with increasing degrees of non-overlap. While the covariate support differs across data sources, the treatment assignments in both RCT and OBS remain well-populated, ensuring that marginal treatment non-overlap (Definition 2.1) does not arise. Figure 3b instead characterizes *joint* distribution mismatch by comparing the empirical conditional treatment probabilities $P(W = w \mid X\text{-bin})$ for $w \in \{0, 1\}$ between RCT and OBS, and quantifying their absolute mass differences across covariate bins. As the latent representations of the two data sources become increasingly aligned, the overlap of their covariate support expands ,from a joint perspective, the discrepancy between conditional treatment probabilities given $X$ diminishes.

### G.5. Baseline Introduction

**Structurally grounded fusion methods.** Early approaches to RCT–OBS integration treat the RCT as the primary source of identification and incorporate observational data through explicit structural assumptions on confounding bias.

- **Experimental Grounding.** Corrects observational estimates using RCTs as an external grounding signal, typically assuming that confounding bias admits a low-complexity parametric representation. While such methods can achieve consistency under limited overlap and correct specification, they rely on strong structural assumptions and become fragile under pronounced covariate shift, where extrapolation is unavoidable.

**Statistical integrative estimators with explicit bias decomposition.** Subsequent work formalizes data fusion by explicitly decomposing observed effects into a true heterogeneous treatment effect (HTE) and a confounding component, establishing identifiability and efficiency gains under transportability and parametric modeling assumptions.

- **Integrative R-Learner.** Extends the R-learner framework to jointly estimate HTEs and confounding functions using both RCT and OBS data. This approach improves efficiency over RCT-only estimation but remains fundamentally limited to regions supported by the RCT and does not address structural or conditional non-overlap.

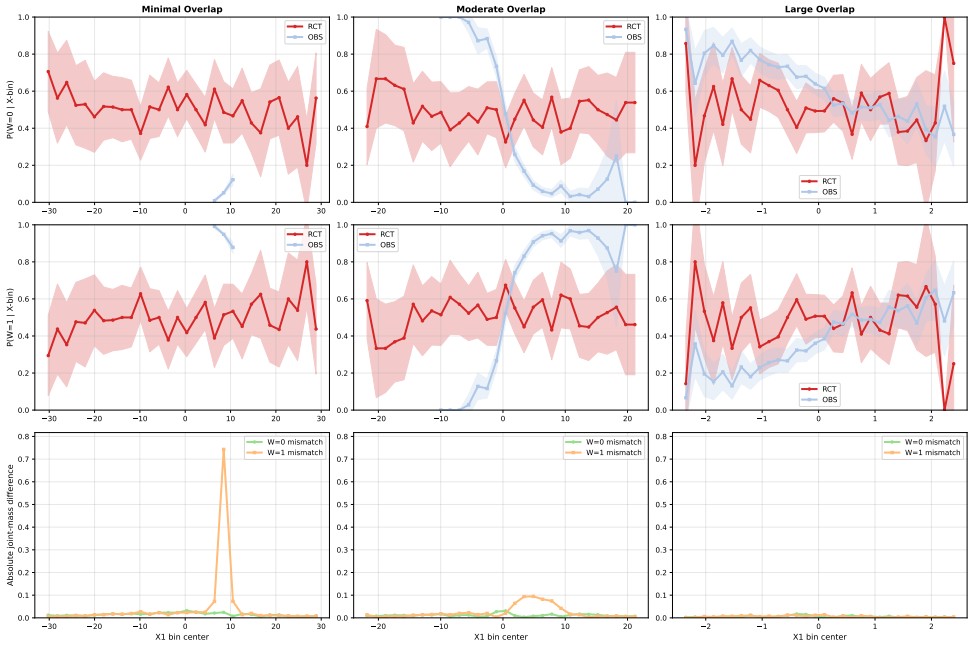

*(a)* Joint Support Mismatch Across Different Overlap Levels

**Support Mismatch Visualization Across Different Overlap Levels**

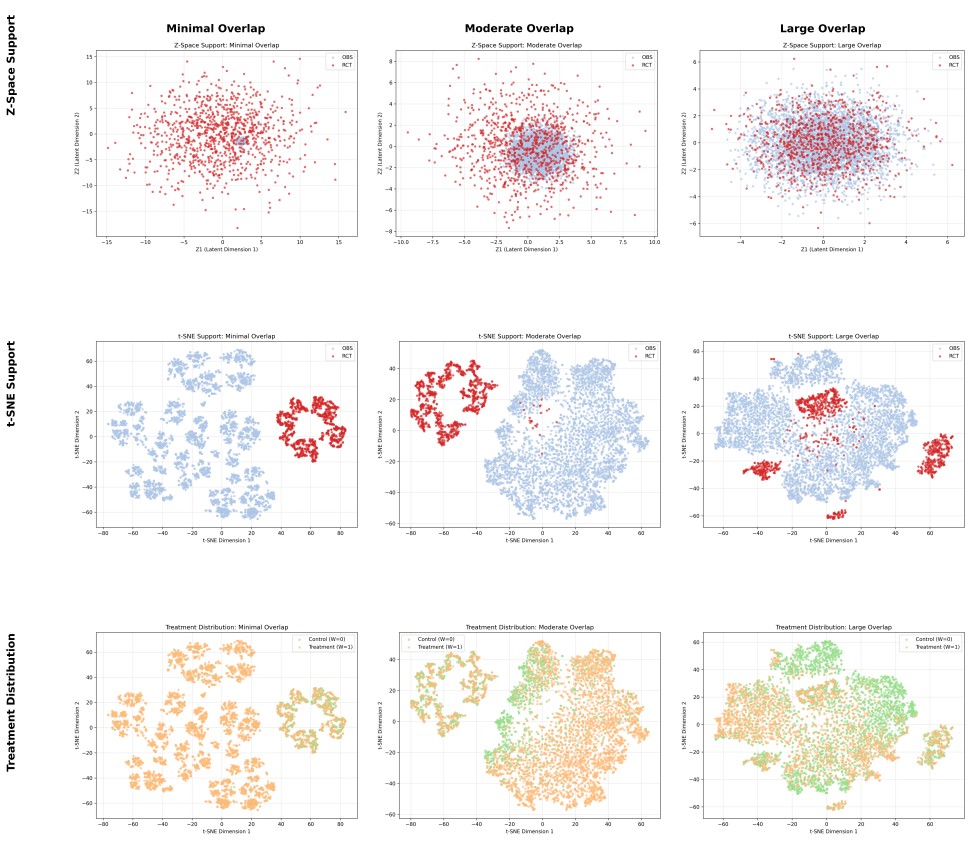

*(b)* Marginal Support Mismatch Across Different Overlap Levels

*Figure 3.* Support Mismatch Across Different Overlap Levels (Two Scenarios)

- **Yang2022 Elastic Integrative Estimator.** Introduces adaptive or elastic combinations of RCT and OBS information to guard against severe observational bias. While improving robustness to misspecification, it focuses on low-dimensional, pre-specified effect modifiers and may discard substantial observational information under strong covariate shift.

- **Yang2024 Integrative HTE Estimator.** Further refines integrative estimation through improved modeling of confounding and efficiency trade-offs. Despite empirical gains, the method continues to rely on overlap assumptions and does not fundamentally resolve conditional support mismatch.

**Orthogonal and RCT-centric estimators.**

- **RCT-only R-Learner.** A Neyman-orthogonal R-learner estimated exclusively on RCT data. This approach guarantees causal identification but ignores observational information, resulting in high variance and limited efficiency when randomized samples are scarce.

**Non-structural neural estimators applied to mixed RCT–OBS data.**    More recent practice often applies flexible neural network–based CATE estimators directly to combined RCT and OBS data, relying on representation learning or balancing regularization to mitigate confounding.

- **T-Learner.** Estimates potential outcomes separately for each treatment using flexible predictors. Although expressive, it is highly sensitive to covariate imbalance and lacks mechanisms to ensure valid data fusion.

- **TARNet.** Learns a shared representation with treatment-specific heads to improve generalization across treatment groups. This approach implicitly assumes observational ignorability and offers no protection against conditional non-overlap.

- **CFRNet.** Augments TARNet with an IPM-based regularizer to align covariate distributions across treatment groups in the latent space. While effective for marginal imbalance, such alignment is insufficient when joint support mismatch arises conditionally on covariates.

- **DragonNet.** Further extends TARNet-style architectures with auxiliary propensity prediction to stabilize training. Despite empirical improvements in some settings, it remains vulnerable to structural non-overlap and lacks identification-based guarantees for RCT–OBS integration.

**Ablations of the proposed method.**    To isolate the contribution of each component, we additionally report two ablated variants of the proposed approach:

- **Own-method-rep** Enforces moment-based feasibility without explicit overlap-aware alignment, limiting robustness under severe conditional non-overlap.

- **Own-method-conv** Performs distribution alignment without enforcing feasibility constraints, leading to inadequate identification when support mismatch is structural.

Overall, these baselines illustrate the progression from structurally grounded fusion methods to flexible but largely non-structural neural estimators.

## H. Sensitivity Analysis

This section provides additional sensitivity analyses for the proposed framework. We first explain the role of key hyperparameters and then examine their effects on feasibility, performance (AUUC), and training stability under varying overlap conditions.

### H.1. Interpretation of Key Hyperparameters

The proposed framework contains several hyperparameters controlling the trade-off among observational fitting, causal validity, and overlap recovery. Recall the augmented Lagrangian objective:

$$\mathcal{L}_\rho(\theta, \phi, \nu) = R_o(\theta, \phi) + \lambda \epsilon_{ov}(\phi) + \langle \nu, g(\theta, \phi) \rangle + \frac{\rho}{2} \|g(\theta, \phi)\|_2^2. \tag{61}$$

**Dual variable ($\nu$).**   The dual variable $\nu$ acts as an adaptive multiplier for randomized moment constraints. It dynamically adjusts the optimization pressure toward satisfying experimental identifying restrictions. Larger values place stronger emphasis on constraint satisfaction, whereas smaller values prioritize observational fitting.

**Augmentation parameter ($\rho$).**   The parameter $\rho$ controls the quadratic penalty term

$$\frac{\rho}{2}\|g(\theta,\phi)\|_2^2,$$

which penalizes residual violations of randomized moments. Unlike $\nu$, which introduces linear constraint enforcement, $\rho$ determines the strength of feasibility regularization. Larger values encourage faster feasibility recovery but excessively large values may introduce optimization instability.

**Wasserstein regularization weight ($\lambda$).**   The Wasserstein weight $\lambda$ controls the contribution of overlap discrepancy

$$\epsilon_{ov}(\phi),$$

which encourages representation learning to reduce distribution mismatch between observational and randomized samples. Small values may fail to adequately recover overlap, whereas overly large values can suppress outcome-relevant information and degrade predictive performance.

**Dual update coefficient.**   The dual update coefficient controls the update magnitude of the dual variable during primal–dual optimization. This parameter determines how rapidly constraint information propagates into optimization. Excessively large values may cause oscillatory behavior, while overly small values slow convergence.

### H.2. Sensitivity and Training Stability

Figures 4–7 report sensitivity analyses under varying overlap conditions. Across a broad range of parameter values, the proposed framework consistently maintains low moment violations while achieving strong AUUC performance. Furthermore, the training curves exhibit smooth convergence behavior, suggesting stable optimization dynamics under different hyperparameter settings.

## I. Sensitivity Analysis

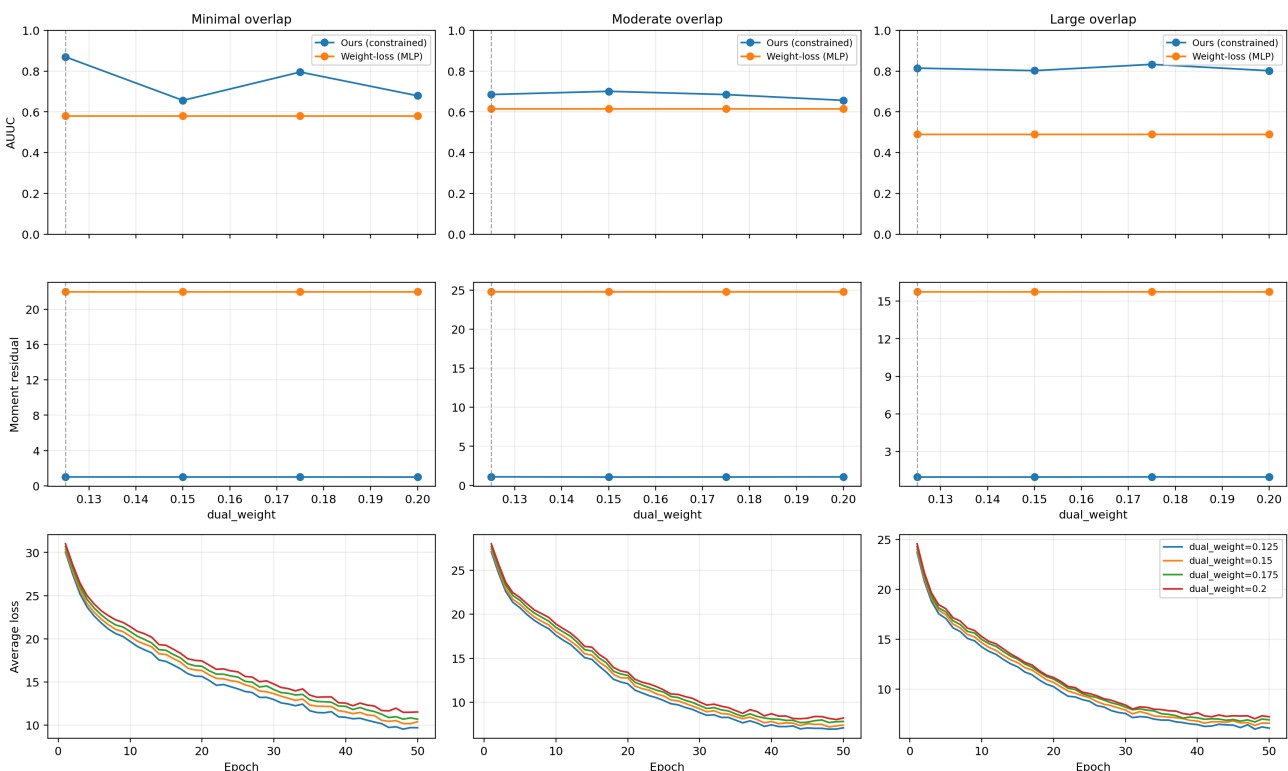

*Figure 4.* Sensitivity analysis with respect to the dual update coefficient.

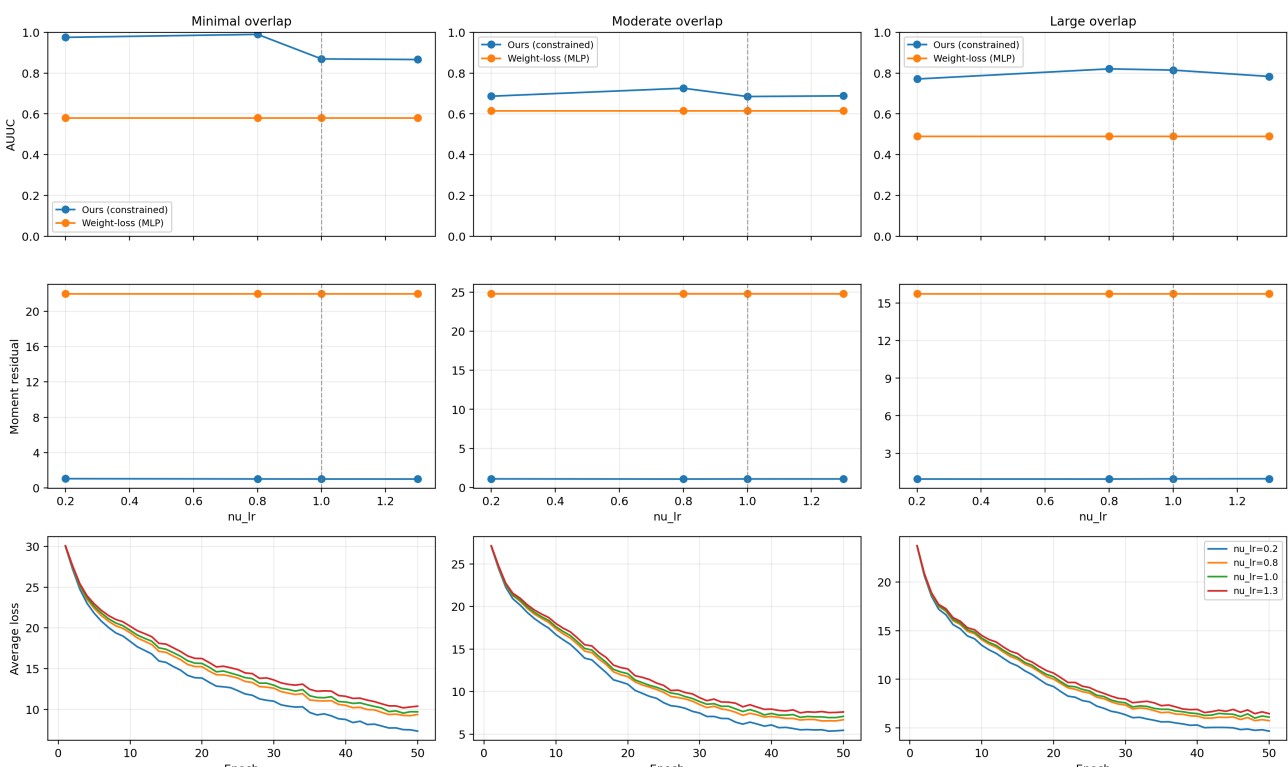

*Figure 5.* Sensitivity analysis with respect to the dual variable $\nu$.

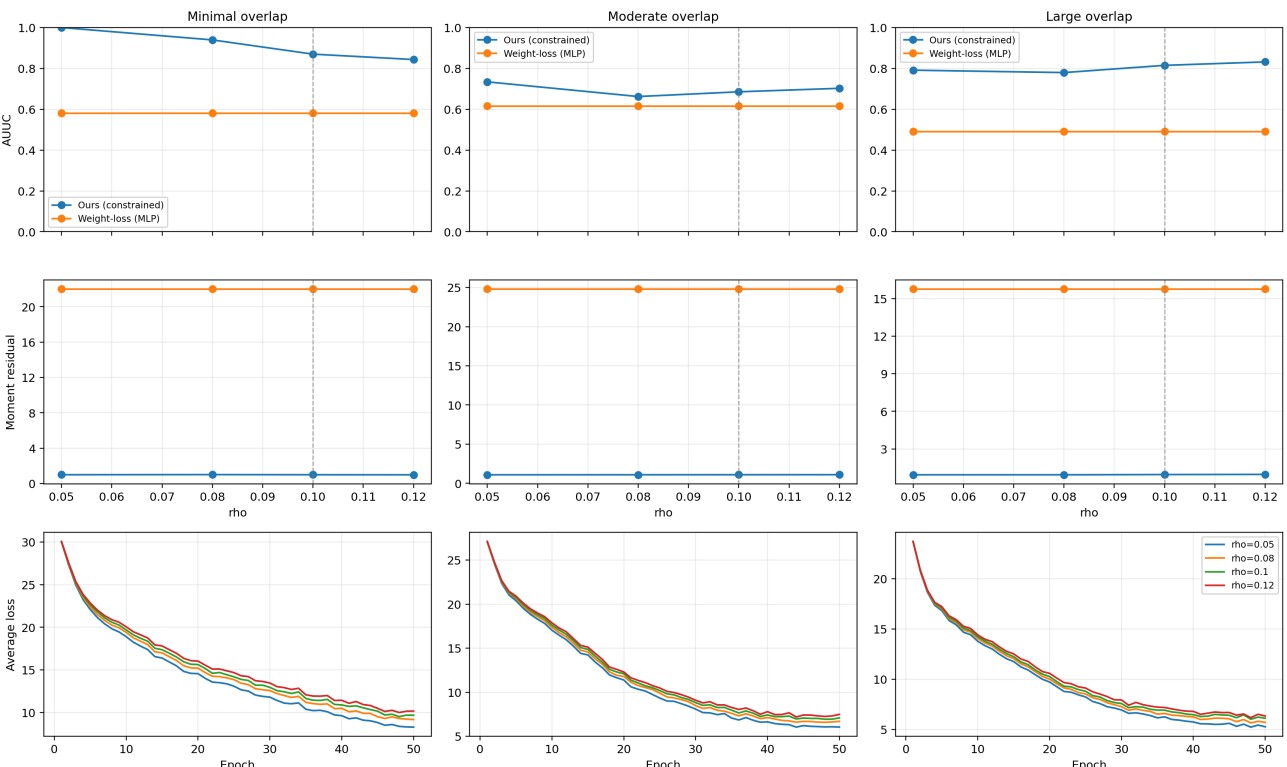

*Figure 6.* Sensitivity analysis with respect to the augmentation parameter $\rho$.

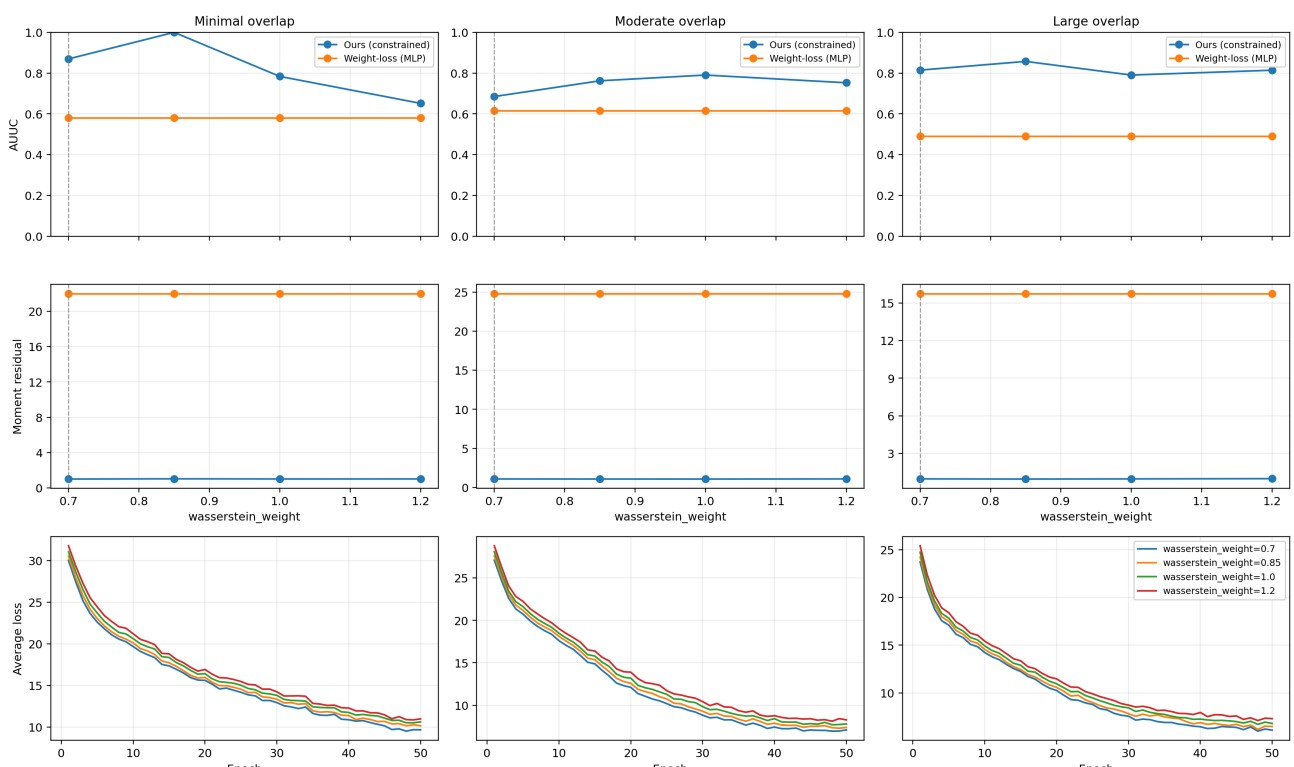

*Figure 7.* Sensitivity analysis with respect to the Wasserstein regularization weight $\lambda$.

