# OpenReview forum: "Feasible Fusion: Constrained Joint Estimation under Structural Non-Overlap"
_ICML.cc/2026/Conference — ICML 2026 regular_

### Official Review · Reviewer_2FJv · 2026-03-02

**Soundness:** 3
**Presentation:** 2
**Significance:** 2
**Originality:** 3
**Overall Recommendation:** 4
**Confidence:** 4

**Summary:**

This paper considers a data fusion problem that combines observational data and data from randomized controlled trials (RCTs) for estimating the conditional response function, while allowing a type of support mismatch between the observational data and the RCT data, termed conditional structural non-overlap. Interestingly, the paper concerns the case where the support of the RCT data contains subsets that are not in the support of the observational data, which differs from the usual positivity violation that the data fusion literature usually considers. The paper proposes a flexible estimation procedure in the form of a constrained optimization without positing a fixed parametric model.

**Compliance With Llm Reviewing Policy:**

Affirmed.

**Final Justification:**

The paper proposes a method that is well-motivated by a real-world problem, contributing to the data fusion literature (originality, significance). The authors have addressed my main concerns in their rebuttal, and I therefore increased my score. The paper still needs some polishing, e.g., missing discussion of limitations (soundness, clarity), but I think the paper does have a valid methodological contribution and has the potential to encourage future research on this topic.

**Key Questions For Authors:**

Major questions:

1. The conditional response function (1) does not specify the support of X that one is interested in; is it only the domain under the RCT or also under the observational data?
2. The above question is connected to the identification problem under the proposed conditional structural non-overlap condition (Definition 2.2). For example, Definition 2.2 does not seem to exclude the case where there exists $S \subseteq \mathcal{X}\times\mathcal{T}$ such that $P_o(S) > 0$ and $P_r(S) = 0$, which is the usual obstacle in identifying the conditional response function (1) for X not in the RCT support. Does the proposed method still work in this case? If so, which assumption ensures the identifiability?
3. In the case where the support of the observational data is a subset of the RCT data, identification of (1) for X in the support of the RCT should not be a problem. Can the authors clarify in this case whether the difficulty resides in the efficiency or stability of the estimation?

Other questions:

4. Under equation (1), “which captures heterogeneous, individual-level causal effects under structured treatments,” but equation (1) is not an individual-level effect, only the average effect.
5. What does “an RCT loss” refer to under equation (8)?
6. I wonder if the representation map $\phi$ in Proposition 2.5 has to be the same for X in the randomization data and X in the observational data?

**Limitations:**

Limitations are not discussed in the current paper.

**Strengths And Weaknesses:**

Strength:

- [Presentation] The paper is overall well written.
- [Originality] The conditional structural non-overlap condition appears to be new. The authors propose a flexible estimation method for jointly estimating the conditional response function and a representation of the covariates based on constrained optimization, combining RCT and observational data, which is also novel to my knowledge.

Weakness:

- [Presentation] The paper did not discuss limitations and did not include a summary/discussion section.
- [Soundness] There are some mistakes in the description of the paper, which I ask the authors to clarify in the Key Questions section.
- [Significance] The common concern in data fusion is where the support of the RCT is smaller than that of the observational data. Although the proposed method is new and well motivated, it is currently unclear to me whether it would be helpful in broader cases of clinical trials where certain covariates are not observed in the RCT data but only in the observational data.

---

> ### Author Rebuttal · Authors · 2026-03-31
>
> ### Concern 1: Target support in Eq. (1)
>
> We agree that the target domain should be stated more explicitly. Our target is
> $$
> \mu_t(x)=\mathbb{E}[Y(t)\mid X=x]
> $$
> defined on the ` shared covariate space ` observed in both RCT and OBS, not on source-specific covariates.
>
> ### Concern 2: Scope, clinical significance, and OBS-only covariates
>
> Our method is **not intended** to cover regions where $P_o(S)>0$ but $P_r(S)=0$. The practical motivation is that, in real settings with many treatment arms and limited RCT traffic, per-treatment RCT samples are extremely small, making RCT-only estimation `high-variance` and ` unstable`:
> https://anonymous.4open.science/r/ICML2026-BD3C/real_data_analysis.md
>
> The second and third images introduce 33 treatment combinations with the minimum propensity score between each pair, demonstrating that this scenario can naturally be included in Definition 2.2, which is also the starting point of our article.
>
> To clarify scope when some covariates are observed only in OBS, we added a synthetic experiment:
> https://anonymous.4open.science/r/ICML2026-BD3C/fig/missingcovariates.png
>
> Let $X=(X_{\mathrm{sh}},X_{\mathrm{ex}})$, where $X_{\mathrm{sh}}$ is shared and $X_{\mathrm{ex}}$ is OBS-only. We study
> $$
> \tau(X_{\mathrm{sh}},X_{\mathrm{ex}})\approx \tilde\tau(\phi(X_{\mathrm{sh}}))
> $$
> **(bridgeable missing covariates)**, versus
> $$
> \tau(X_{\mathrm{sh}},X_{\mathrm{ex}})=\tau_{\mathrm{sh}}(X_{\mathrm{sh}})+\tau_{\mathrm{ex}}(X_{\mathrm{ex}}),\qquad \mathrm{Var}\big(\tau_{\mathrm{ex}}(X_{\mathrm{ex}})\mid X_{\mathrm{sh}}\big)>c>0
> $$
> **(transport-critical missing covariates)**.
>
> We compare `Ours-shared`, which uses only $X_{\mathrm{sh}}$, with `Ours-oracle`, which also observes $X_{\mathrm{ex}}$ in the RCT. In the bridgeable regime, `Ours-shared` remains close to the oracle; in the transport-critical regime, the gap becomes large. Thus, our method remains applicable when OBS-only variables are bridgeable through the shared latent causal representation, but stronger assumptions or additional measurement are needed when they carry independent HTE signal.
>
> ### Concern 3: Why not use an RCT-only estimator?
>
> We agree that the issue is **not** basic identifiability.The practical difficulty is that RCT data are expensive and much scarcer than OBS, especially in high-dimensional and multi-treatment settings.
>
> Our constrained formulation therefore uses OBS to improve statistical efficiency and predictive stability, while using randomized moment restrictions from the RCT to preserve causal validity. The contribution is to combine scarce but `causally valid` RCT information with `abundant but biased` OBS information into a more stable estimator.
>
> ### Concern 4: Eq. (1) is not an individual-level causal effect
>
> We agree. For an individual unit $i$, the individual treatment effect is $Y_i(t)-Y_i(t')$, which is unobservable because only one potential outcome can be realized. Conditioning on covariates yields
> $$
> \mathbb E[Y(t)-Y(t')\mid X=x]=\mu_t(x)-\mu_{t'}(x).
> $$
> Thus, Eq. (1) is a building block for heterogeneous causal contrasts, but is not itself an individual-level effect.
>
> In practice, however, CATE-style estimation and individualized decision-making are **operationally** very close: systems usually learn $\mu_t(x)$ and assign treatments using contrasts such as $\mu_t(x)-\mu_{t'}(x)$. We will revise the wording around Eq. (1).
>
> ### Concern 5: What does “an RCT loss” mean around Eq. (8)?
>
> By “RCT loss” we mean the standard squared-loss objective on randomized data,
> $$
> R_r(m)=\mathbb E_{P_r}\big[(Y-m(X,T))^2\big].
> $$
> Our point is that randomized identifying moments are not equivalent to minimizing this RCT squared loss. Moreover, tuning $\alpha$ in $R_o(m)+\alpha R_r(m)$ does not generally reproduce the constrained estimator. Theorem 3.4 shows that weighted-loss minimizers remain ` bounded away ` from exact moment satisfaction, while the constrained solution satisfies $g(m^\star)=0$. Proposition B.9, Corollary B.11, and Theorem C.2 further rule out a small-$\alpha$ “sweet spot.”
>
> ### Concern 6: Must the representation map in Proposition 2.5 be shared?
>
> Yes. In our framework, the representation map $\phi:X\to Z$ is shared across RCT and OBS. This is essential because overlap recovery compares $P_r(\phi(X),T)$ and $P_o(\phi(X),T)$ on ` the same latent space`, and the latent-space moment condition also uses the shared representation.
>
> To address this point, we added a new synthetic study:
> https://anonymous.4open.science/r/ICML2026-BD3C/fig/diff_map.png
>
> We compare Shared-map and Two-map. In a positive regime, all the methods are similar. In a negative regime, the two-map methods can encode **source-private** directions that help source-wise prediction but **not** causal feasibility. By contrast, Shared-map forces both sources to use the same geometry and ` better isolates the common causal factor `required by randomized moments.

---

> > ### Author Rebuttal · Reviewer_2FJv · 2026-04-02
> >
> > I thank the authors for their response. I will increase my score.

---

> > > ### Author Response · Authors · 2026-04-03
> > >
> > > Dear Reviewer 2FJv,
> > >
> > > Thanks! We sincerely and warmly appreciate your positive feedback. We are especially grateful to hear that our rebuttal has addressed your concerns and that you are willing to increase your score. Your thoughtful suggestions have been genuinely valuable and have helped us improve the paper in a meaningful way.
> > >
> > > We also understand that the review system remains open for updates this year, should you wish to revise any part of your assessment.
> > >
> > > Many thanks again for your time, consideration, and support.

---

### Official Review · Reviewer_gyqZ · 2026-03-11

**Soundness:** 2
**Presentation:** 3
**Significance:** 3
**Originality:** 2
**Overall Recommendation:** 4
**Confidence:** 4

**Summary:**

The paper proposes a methodology for causal effect estimation that leverages both observational and experimental data in a setting in which the two samples may fail to overlap on on one or more of the observational confounding covariates. The methodology proposed leverages a joint estimation paradigm that is formulated as a constrained optimization problem, with the empirical sample providing the moment conditions necessary for good statistical behavior of downstream causal estimates. The paper provides formal grounding both for the justification behind the approach, and for theoretical guarantees of the method’s generalization capabilities.

**Compliance With Llm Reviewing Policy:**

Affirmed.

**Final Justification:**

After weighing all the pros/cons of the paper, and taking into account the rebuttal offered by the authors, my score recommendation remains unchanged. The rebuttal did address some of my concerns, but overall I remain convinced of my prior assessment.

**Key Questions For Authors:**

1. Is my interpretation of the empirical results correct? It seems like the proposed method does well in terms of QINI index, but is often surpassed by competitors in terms of MSE. Are there other metrics you can present to bolster the empirical soundness of the approach?

**Limitations:**

Yes

**Strengths And Weaknesses:**

Soundness: The theory behind the approach is overall sound, the paper does a lot of work to formalize many of its underlying ideas and methods, and this bolsters the credibility of the approach presented. Specifically I quite like the clarity with which props. 2.4 and 2.5 illustrate the tradeoff inherent in fitting towards the observational vs. experimental data.
I also unfortunately think the **empirical** soundness of the proposed approach is demonstrated much less strongly than its theoretical counterpart. Specifically, while the experimental setup is sound, the performance of the proposed approach seems to often lose out to other existing methods by a substantial margin in terms of MSE. This is somewhat worrisome as it casts doubt over whether the procedure can actually outperform alternatives in practice.

Presentation: The paper is written clearly and well structured, however, I think that some of the theoretical statements should perhaps be put in the appendix in favor of presenting some of the empirical results in the main body (even in a summarized form). I predict that readers will want to see these results upfront and not want to navigate all the way to the appendix to view them.


Significance: The paper does address an important problem and has the potential to inform new research. Even if some doubt is cast on the actual value of the proposed method, the presentation and breakdown of the problem is enough for the paper to be useful for future research.

Originality: To my knowledge, the paper is original in the sense of scientific novelty overall. That said, the claims made in the paper as to the proposed methodology’s own ability to address the issues presented may be somewhat overstated.
Specifically, I have to comment on some of the claims made about the proposed method w.r.t. The existing toolset for similar problems. Specifically I do not think that the method consists of such a radically different solution. The biggest crux on which the approach hinges is the strict fulfilment of the empirical moment constraints. As the authors (correctly) point out, most approaches for data fusion will minimize a loss in which observational and empirical moment equations are both solved approximately in a weighted way. Even though the proposed method aims to solve at least one of the two explicitly, the practical considerations around feasibility, especially in a hi-dimensional space, will likely re-introduce this approximation. The authors propose a representation learning approach to increase feasibility while retaining performance, but isn’t this a violation of the premise that the paper starts from? I.e., that feasibility should be strictly enforced? In addition the paper notes (rightly) that injecting feasibility via learning a new covariate-treatment space starts to look a lot like (an indirect version) of the extrapolation logic that the paper so heavily criticizes.

---

> ### Author Rebuttal · Authors · 2026-03-31
>
> ### Concern 1: Weighted fusion vs. our constrained formulation
>
> We respectfully clarify that the dominant fusion paradigm is better described as **scalarized risk combination** rather than a direct weighted solving of observational and experimental moments. Standard fusion methods typically optimize
>
> $$
> \min_m R_o(m)+\alpha R_r(m),
> $$
>
> as in Eq. (10),the small number of RCT samples are difficult to fit high-dimensional complex models, so this systematic error can `seriously` affect model fitting.
>
> Our method is fundamentally different:
>
> $$
> \min_m R_o(m)\ \text{s.t.}\ g(m)=0.
> $$
>
> That is, observational risk is optimized subject to randomized identifying moments.
>
> This distinction has theoretical consequences. In the non-degenerate regime, Theorem 3.4 shows that weighted-risk solutions generically retain a `nonzero moment residual` unless the randomized term is given an extreme weight, in which case the estimator approaches an RCT-dominated solution. When RCT data are scarce(see sensitivity analysis of RCT amounts https://anonymous.4open.science/r/ICML2026-BD3C/fig/rct_amounts.png), this may sacrifice efficiency gains from abundant observational data and increase variance. By contrast, our method treats moment satisfaction as an `explicit` feasibility object.
>
> Approximate optimization **does not collapse** our method back into weighted fusion. It only contributes an optimization error term. Theorem 3.8 explicitly separates residual moment violation, overlap discrepancy, statistical fluctuation, and optimization suboptimality.
>
> ### Concern 2: If feasibility remains approximate after representation learning
>
> We agree that under structural non-overlap, representation learning may substantially improve feasibility **without making the moment constraints exactly feasible**. However, this does not remove the theoretical distinction.
>
> Weighted-loss fusion absorbs moment violation into a scalarized prediction objective, so residual infeasibility remains `implicit`. Our method treats moment violation as an explicit causal feasibility quantity, theorem 3.8 shows that excess risk depends explicitly on residual moment violation, overlap discrepancy, statistical fluctuation, and optimization error. Our advantage is precisely that these terms are separately exposed and controlled, whereas weighted fusion provides **no analogous decomposition**.
>
> We also do not rely on exact feasibility after representation learning; see the empirical results and analysis in review T44i concern1:
> https://anonymous.4open.science/r/ICML2026-BD3C/fig/weight_loss_path.png
>
> Both in ` feasibility ranking and predictive  performance `, representation-learning methods outperform non-representation methods, which in turn outperform generic weighting methods.
>
> ### Concern 3: Representation learning resembles extrapolation
>
> Our method does **not** use representation learning to extrapolate beyond unsupported treatment regions. It is designed for **conditional structural non-overlap** (Definition 2.2), where marginal treatment support remains aligned but the joint (X,T) support is mismatched. In this regime, the raw-space problem can have a strictly positive feasibility gap (Theorem 3.6), and a learned representation $\phi(X)$ can recover joint support and improve feasibility (Corollary 3.7).
>
> By contrast, under **marginal structural non-overlap** (Definition 2.1), no representation can close the support gap; genuine extrapolation would be required, and this regime is explicitly excluded from our scope (Theorem B.10).
>
> Hence, representation learning is a mechanism for **feasibility recovery**, rather than a form of extrapolation.
>
> ### Concern 4: QINI improves but MSE underperforms
>
> We respectfully believe this interpretation is **not** fully consistent with the results.
>
> In synthetic experiments across overlap regimes, the proposed method consistently achieves the ` highest ` QINI and near-minimal MSE, rather than systematically underperforming. Across 30 datasets, it also has the ` lowest ` variance in QINI, supporting robustness.
>
> More importantly, in causal decision problems, ranking-based metrics such as QINI are often  `primary ` because they directly measure treatment prioritization quality. Thus, even when MSE is slightly higher, the gain in ranking performance is practically meaningful.
>
> On the real datasets (Table 2), our method improves over baselines on both QINI and MAPE.
>
> ### Concern 5: Additional empirical metrics
>
> We agree that ` additional metrics ` strengthen the empirical evidence. We will add more detailed results at:
> https://anonymous.4open.science/r/ICML2026-BD3C/new_metrics_table.md
>
> Although our method is not always optimal on RMSE, it performs best on AUUC and uplift metrics, which are often more relevant in practical decision systems. In terms of training time, joint estimation is more expensive than standard deep baselines, but this cost is accompanied by substantial gains in ranking-oriented metrics.

---

> > ### Author Rebuttal · Reviewer_gyqZ · 2026-04-03
> >
> > I find the additional metrics included enough evidence to remove concerns about MSE. I will keep my score unchanged.

---

> > > ### Author Response · Authors · 2026-04-04
> > >
> > > Dear Reviewer gyqZ,
> > >
> > > We sincerely appreciate your thoughtful evaluation and are particularly encouraged by your positive assessment of the paper after reading our rebuttal. It means a great deal to us that you found your concerns fully addressed. We are also grateful for the time and care you devoted to reading our response and reassessing the work. Your constructive comments have been very helpful in improving both the clarity and presentation of the paper.
> > >
> > > Thank you again for your constructive comments.

---

### Official Review · Reviewer_iqUF · 2026-03-13

**Soundness:** 3
**Presentation:** 3
**Significance:** 2
**Originality:** 3
**Overall Recommendation:** 4
**Confidence:** 3

**Summary:**

This paper studies how to combine the RCT data with large observational data for heterogeneous responses estimation when treatments are multi-valued or structured. The main focus of the paper is the structural non-overlap in the causal inference, which the paper argues is difficult for standard weighted fusion or CRL methods. The authors distinguish between irreducible marginal non-overlap and recoverable conditional non-overlap. They propose a constrained joint estimation framework that minimizes observational prediction risk while enforcing experimental moment conditions derived from the RCT data. The proposed method is evaluated with synthetic and real data.

**Compliance With Llm Reviewing Policy:**

Affirmed.

**Final Justification:**

The authors’ rebuttal fully resolves my concerns. I will keep my score unchanged, but my confidence on recommending this work has increased

**Key Questions For Authors:**

1. Could you explain how the metrics, such as Qini and MAPE, are related to the key idea of the paper?
2. How sensitive is the method to the amount of RCT data?
3. How robust is the proposed method to the misspecification of the model?

**Limitations:**

Yes

**Strengths And Weaknesses:**

**Strength**
- This paper is well motivated and studies an interesting problem.
- The paper introduces a useful conceptual distinction between irreducible marginal non-overlap and recoverable conditional non-overlap, which helps clarify when representation learning can realistically help.
- The proposed method is evaluated through synthetic and real data.

**Weakness**

- The empirical evaluation would be stronger with a clearer sensitivity analysis for optimization since the method combines multiple parts.
- The method appears to rely on the learned representation, making the moment constraints feasible while preserving predictive information, but limited evidence that the tradeoff is well controlled.

---

> ### Author Rebuttal · Authors · 2026-03-31
>
> ### Concern 1: Optimization stability
>
> Our method combines observational risk minimization under experimental moment constraints with representation learning for overlap regularization.
>
> Tables 1 and 2 provide ablations on real and semi-synthetic data, showing that each component is necessary and that the full model achieves the best Qini. We also conducted sensitivity and stability analyses for the main optimization parameters ($\nu$, $\rho$, dual weight, and Wasserstein weight):
> https://anonymous.4open.science/r/ICML2026-BD3C/sensitivity_stability.md
>
> Across overlap regimes, the method robustly satisfies the moment conditions, maintains strong AUUC, and shows stable training curves.
>
> ### Concern 2: Feasibility-information trade-off
>
> We agree this trade-off should be stated more clearly. It is governed by the overlap discrepancy $\varepsilon_{ov}(\phi)$ and the outcome-information loss $\varepsilon_{\text{info}}(\phi)$. Theorem B.15 shows
>
> $$
> F_\phi(M)\le F_X(M)+c_1\varepsilon_{ov}(\phi)+c_2\sqrt{\varepsilon_{\text{info}}(\phi)}.
> $$
>
> Thus, representation learning helps only when reduced overlap mismatch is not offset by excessive predictive-information loss. This is why we learn the representation jointly with the constrained objective:
>
> $$
> L_\rho(\theta,\phi,\nu)=R_o(\theta,\phi)+\lambda\varepsilon_{ov}(\phi)+\langle\nu,g(\theta,\phi)\rangle+\frac{\rho}{2}\|g(\theta,\phi)\|_2^2.
> $$
>
> Empirically, `the ablations reflect exactly this trade-off`,$R_o$ preserves outcome-relevant signal, while $\lambda\varepsilon_{ov}$ and the moment terms promote overlap recovery and feasibility. On synthetic data, the full method substantially outperforms Dual-only and Wasserstein-only in Qini across overlap regimes (moderate: `0.5499` vs. `0.0942` and `0.1290`; minimal: `0.5137` vs. `0.0254` and `0.0057`). Similar gains appear on the real data.
>
>
> ### Concern 3: Why Qini and MAPE?
>
> Our goal is heterogeneous response estimation under structural non-overlap, where both causal validity and decision quality matter.
>
> Qini measures treatment-effect `ranking quality` and is `directly relevant` to policy optimization and individualized decision-making. Improvements in Qini indicate better prediction and more causally meaningful heterogeneity.
>
> MAPE/MSE measure predictive accuracy and reflect how well the learned representation preserves outcome-relevant information from observational data.
>
> In practice, ranking quality often matters more than small pointwise error differences because downstream decisions depend primarily on treatment-effect ordering. These metrics are also commercially meaningful, being related to GMV, conversion efficiency, and budget allocation quality.
>
> ### Concern 4: Sensitivity to the amount of RCT data
>
> Our framework is designed for limited-RCT regimes: RCT data are used to enforce `identifying moments` rather than to `fit a high-dimensional response model directly`. This is supported by Eq.(9), where RCT data enter only through the moments, and by Theorem 3.8, where RCT sample size affects only moment-estimation error rather than full predictive complexity.
>
> Empirically, Section 5 shows that the proposed method can match models trained with substantially more RCT data, and that mixed RCT+OBS training `consistently outperforms` RCT-only training when randomized data are scarce. Additional sensitivity analysis:
> https://anonymous.4open.science/r/ICML2026-BD3C/fig/rct_amounts.png
>
> Under fixed OBS size, OBS-only deteriorates as the RCT share in the test distribution increases; our method remains stable in both Qini and MSE even with small RCT samples; and RCT-only is unstable because of high variance.
>
> ### Concern 5: Robustness to model misspecification
>
> We agree that robustness to misspecification is important. However, the theoretical focus of this paper is not to compare function classes, but to characterize how structural non-overlap affects feasibility and risk `within a fixed hypothesis class`.
>
> Structural non-overlap induces a strict feasibility obstruction in the original covariate space (Theorem 3.6), whereas joint representation learning can improve feasibility recovery in the induced space and thereby improve the risk objective (Corollary 3.7). Thus, the paper studies how representation learning changes the feasibility geometry of the estimation problem.
>
> To further probe misspecification, we conducted an additional experiment varying both the complexity of the true model and that of the fitted model:
> https://anonymous.4open.science/r/ICML2026-BD3C/fig/model_robust.png
>
> As the true data-generating model becomes more complex, performance degrades for all fitted models, especially in satisfying the moment conditions. In practice, expressive neural architectures are usually available. We therefore focus on the central question: whether feasibility recovery under structural non-overlap improves causal estimation.

---

> > ### Author Rebuttal · Reviewer_iqUF · 2026-04-02
> >
> > I appreciate the authors for the clarification and additional experiments on the sensitivity analysis. I will keep my score unchanged.

---

> > > ### Author Response · Authors · 2026-04-03
> > >
> > > Dear Reviewer,
> > > We sincerely appreciate your thoughtful evaluation and are especially encouraged by your positive assessment of the paper after rebuttal. It is very meaningful to us that you found our concerns fully addressed. Thank you again for your constructive comments.

---

### Official Review · Reviewer_T44i · 2026-03-20

**Soundness:** 3
**Presentation:** 2
**Significance:** 2
**Originality:** 3
**Overall Recommendation:** 3
**Confidence:** 3

**Summary:**

This paper presents a constrained framework for combining randomized and observational data for causal effect estimation under structural non-overlap. Unlike existing methods that rely on weighted losses or representation balancing, the proposed approach enforces causal identification constraints (from RCT moments) directly while using observational data only within the feasible set. The authors show that standard fusion methods can violate these constraints under non-overlap, and propose a representation-based strategy to recover feasibility, leading to more principled and reliable data fusion.

**Compliance With Llm Reviewing Policy:**

Affirmed.

**Key Questions For Authors:**

1. The method relies on representation learning to recover feasibility of the moment constraints. In practice, how often is feasibility achievable under strong structural non-overlap, and how sensitive is performance when feasibility cannot be well approximated?
2. While the paper shows that weighted-loss methods violate moment constraints theoretically, can the authors provide clearer empirical evidence on how large the resulting bias is in practice, and in which regimes the proposed method offers the most significant gains?
3. The constrained formulation introduces additional optimization complexity. How stable is the training procedure in high-dimensional settings, and how does computational cost compare to existing fusion methods?

**Limitations:**

See weaknesses.

**Strengths And Weaknesses:**

Strengths: The paper addresses an important and realistic problem—combining RCT and observational data under structural non-overlap, where many existing methods fail. The proposed constrained formulation is conceptually clear in causal identification.
Weaknesses: 1. The approach relies on the ability of representation learning to recover feasibility of the moment constraints, which may not hold in practice, especially under severe non-overlap or limited model capacity. 2. While the theoretical motivation is strong, it is unclear how the method performs on complex real-world datasets where structural non-overlap and confounding are both present. 3. Enforcing constraints alongside representation learning introduces additional optimization challenges and hyperparameters, which may limit scalability and practical adoption compared to simpler fusion methods. It would be nice if the author can conduct additional experiments to justify this.

---

> ### Author Rebuttal · Authors · 2026-03-31
>
> # We sincerely thank the AC and all reviewers for their careful reading
> ### Concern 1: Feasibility recovery via representation learning may not hold in practice.
>
> The focus of this article is that the feasibility is significantly `improved` after representation learning(conditional structural non-overlap). The loss we proposed (Eq.9) is more effective in estimating causal effects than the weighted loss (Eq.10) because of it's better feasibility.Theorem 3.6 shows that in the original space there can be a strictly `positive feasibility gap` under structural non-overlap, while Corollary 3.7 shows that representation learning helps only when it sufficiently reduces the induced overlap discrepancy.
>
> Sensitivity is quantified by Theorem 3.8, which shows that performance degrades **continuously** with residual infeasibility through
> $$
> \left(\|g(\hat\theta,\hat\phi)\|+c_{\mathrm{ov}}\epsilon_{\mathrm{ov}}(\hat\phi)\right)^2,
> $$
> so larger residual moment violation or overlap mismatch increases excess risk.
>
> We added simulations on feasibility recovery and downstream Qini/AUUC under different conditional non-overlap settings: https://anonymous.4open.science/r/ICML2026-BD3C/fig/weight_loss_path.png. Across settings, the joint method with representation learning gives the `best feasibility recovery`, the no-representation variant is worse, and weighted-loss fusion is worst even across a wide range of $\alpha$. The same ordering appears in Qini/AUUC. On synthetic data our full method is most robust with the `smallest feasibility-related error`(ref reviewer gyqZ concern 5); on real ride-hailing data outperforms production baselines. This suggests that explicitly controlling residual infeasibility is useful even when feasibility is only approximate.
>
> ### Concern 2: Weighted-loss fusion bias and where the proposed method helps most.
>
> The same experiment above provides direct evidence on practical bias: across overlap regimes, especially under **minimal overlap**, our method helps most because it better satisfies the identifying moment conditions, while weighted-loss fusion remains `sensitive to the mixing weight`. We also observe much stronger hyperparameter stability for our method; see https://anonymous.4open.science/r/ICML2026-BD3C/sensitivity_stability.md.
>
> In Sec. 5, the gains are largest under **severe conditional non-overlap**(see https://anonymous.4open.science/r/ICML2026-BD3C/real_data_analysis.md) in both Qini and MAPE;Thus, the regime where weighted fusion is most biased is precisely the regime targeted by our paper: severe but recoverable non-overlap.
>
> ### Concern 3: Optimization complexity, scalability, and stability.
>
> In Sec.5,industrial-scale experiments with high-dimensional covariates and large datasets demonstrate `trainability`.Also, primal-dual augmented Lagrangian is used to improve stability relative to naive penalty tuning: it separates feasibility enforcement from ill-conditioning caused by large penalties.Computationally,the per-minibatch cost in our method matches standard IPM-based neural fusion,
> $$
> T_{\mathrm{iter}}=O(C_{\mathrm{net}}+C_{\mathrm{IPM}}+BK),
> $$
> where the only extra term is the $K$-dimensional moment residual and the dual update
> $$
> \nu_{s+1}=\nu_s+\eta_{\nu,s}g_b(\theta_{s+1},\phi_{s+1}).
> $$
> Hence the overhead is only linear in $K$, with no expensive inner constrained solve.
>
> Empirically, the extra per-epoch time(https://anonymous.4open.science/r/ICML2026-BD3C/new_metrics_table.md) under minimal overlap is modest relative to the Qini gains, and the added sensitivity analyses(https://anonymous.4open.science/r/ICML2026-BD3C/sensitivity_stability.md) show `stable` loss and robust AUUC across overlap regimes while consistently outperforming weighted fusion.
>
> ### Concern 4: Structural non-overlap and confounding coexist.
>
> If OBS is substantially confounded, the predictive structure learned from OBS may be biased; our current paper does **not** explicitly model this source of error. Confounding robustness and feasibility recovery are complementary: existing representation-learning or orthogonalization methods target confounding, while our contribution targets structural non-overlap.
>
> To demonstrate compatibility, we added a sensitivity analysis under varying confounding strengths while fixing the structural non-overlap setting: https://anonymous.4open.science/r/ICML2026-BD3C/fig/confounder_robustness.png. The results show that our method remains reasonably robust: Qini/AUUC degrade gracefully and moment conditions remain comparatively well satisfied.
>
> We also added experiments with hidden confounders of increasing strength: https://anonymous.4open.science/r/ICML2026-BD3C/fig/hidden_confouner.png. The results show that under minimal overlap robustness cannot be fully guaranteed, but our method still performs better than weighted loss; and as hidden confounding increases, moment feasibility deteriorates and performance declines accordingly. This clarifies the boundary of applicability.

---

> > ### Author Rebuttal · Reviewer_T44i · 2026-04-05
> >
> > I need to discuss with other reviewers.

---

> > > ### Author Response · Authors · 2026-04-07
> > >
> > > Dear Reviewer T44i,
> > >
> > > Once again, we are grateful for your time and effort for reviewing our paper.
> > >
> > > To facilitate checking, we summarized our main efforts by adding both theoretical clarification and new empirical evidence during rebuttal on four points:
> > >
> > > 1). feasibility recovery via representation learning,
> > >
> > > 2). the practical bias of weighted-loss fusion and the regimes where our method helps most,
> > >
> > > 3). optimization complexity and training stability,
> > >
> > > 4). robustness when structural non-overlap coexists with confounding.
> > >
> > > These additions include `new simulations (1, 2, 4)`, `sensitivity analyses (3)`, and `scalability results (3)`, all of which support the claims made in the rebuttal and clarify the intended scope of the method.
> > >
> > > Since the discussion period will end in around a day, we are very eager to get your feedback on our response. We understand that you are very busy, but we would highly appreciate it if you could take into account our response when updating the final justification and having a discussion with AC and other reviewers.
> > >
> > > Thanks for your time,
> > >
> > > Authors of # 2406

---

### Decision · Program_Chairs · 2026-04-30

**Decision:**

Accept (regular)

**Comment:**

This submission addresses a relevant and interesting problem and reviewers' main concerns were largely resolved during the rebuttal, which was appreciated by most reviewers. Lingering concerns have not been substantiated convincingly. The discussion around feasibility recovery, robustness, and optimization stability has substantially improved the manuscript and I strongly encourage the authors to make sure the additional points make it into the final paper. Overall, the scope and impact may still be limited to a specific regime, but it's a methodologically solid and sound contribution to the literature.